# Re-engagement in care of people living with HIV lost to follow-up after initiation of antiretroviral therapy in Mali: Who returns to care?

Aliou Baldé[1]*, Laurence Lièvre[1], Almoustapha Issiaka Maiga[2], Fodié Diallo[3], Issouf Alassane Maiga[4], Dominique Costagliola[1], Sophie Abgrall[1,5]

1 Sorbonne Université, INSERM, Institut Pierre Louis d'Epidémiologie et de Santé Publique, IPLESP, Paris, France, 2 Unité d'épidémiologie moléculaire de la résistance du VIH aux ARV du Centre de Recherche et de Formation sur le VIH/SIDA et la tuberculose (SEREFO), Université des Sciences, des Techniques et des Technologies de Bamako (USTTB), Bamako, Mali, 3 Association de Recherche, de Communication, d'Accompagnement à Domicile des Personnes Vivant avec le VIH (ARCAD-SIDA), Bamako, Mali, 4 Ensemble pour une Solidarité Thérapeutique Hospitalière en Réseau (ESTHER)/Expertise France, Bamako, Mali, 5 AP-HP, Hôpital Antoine Béclère, Service de Médecine Interne, Clamart, INSERM, Université Paris Sud, Université Paris Saclay, France

* aliou.balde@iplesp.upmc.fr

## Abstract

### Objectives

We assessed cumulative incidence rates of and factors associated with re-engagement in HIV care for PLHIV lost to follow-up in Mali.

### Methods

HIV-1-infected individuals lost to follow-up before 31/12/2013, $\geq$ 18 years old, who started ART from 2006 to 2012 at one of 16 care centres were considered. Loss to follow-up (LTFU) was defined as an interruption of $\geq$ 6 months during follow-up. The re-engagement in care in PLHIV lost to follow-up before 31/12/2013 was defined as having at least one clinical visit after LTFU. The cumulative incidence rates of re-engagement in care was estimated by Kaplan-Meier and its predictive factors were assessed using Cox models. Socio-demographic characteristics, clinical and immune status, period, region, centre expertise level, and distance from home at the start of ART plus a combined variable of duration of ART until LTFU and 12-month change in CD4 count were assessed. Multiple imputation was used to deal with missing data.

### Results

We included 3,650 PLHIV lost to follow-up before December 2013, starting ART in nine outpatient clinics and seven hospitals (5+2 in Bamako and 4+5 in other regions): 35% male, median (IQR) age 35 (29–43), and duration of ART until LTFU 11 months (5–22). Among these PLHIV, 1,975 (54%) were definitively LTFU and 1,675 (46%) subsequently returned to care. The cumulative incidence rates of re-engagement in care rose from 39.0% at one

**Data Availability Statement:** All relevant data are within the manuscript and its Supporting Information files.

**Funding:** The author(s) received no specific funding for this work.

**Competing interests:** The authors have declared that no competing interests exist.

year to 47.0% at three years after LTFU. Predictors of re-engagement in care were starting ART with WHO stage 1–2 and CD4 counts $\geq$ 200 cells/μL, being treated for $\geq$ 12 months with CD4 count gain $\geq$ 50 cells/μL, or being followed in Bamako. People followed at regional hospitals or outpatient clinics $\geq$ 5 km away, or being treated for $\geq$ 12 months with CD4 count gain < 50 cells/μL were less likely to return to care.

## Conclusions

Starting ART with a higher CD4 count, better gain in CD4 count, and being followed either in Bamako or close to home in the regions were associated with re-engagement in care.

## Introduction

The discontinuation of antiretroviral therapy (ART) due to loss to follow-up (LTFU) is still a big challenge for ART programmes in sub-Saharan Africa and is an obstacle for attainment of the second and third targets of the UNAIDS 90-90-90 HIV treatment targets (diagnosis of 90% of HIV-infected people, provision of treatment for 90% of people diagnosed with HIV, and viral undetectability in 90% of treated people) as it affects the sustainable treatment and virological success of people living with HIV (PLHIV) on ART [1, 2]. So, identifying and re-engaging PLHIV lost to follow-up is necessary [3].

The definition of LTFU varies widely across programmes and countries [2]. However, the absence of a clinical visit at least six months before database closure, which corresponds to at least two missed appointments for PLHIV who are on a quarterly appointment schedule, is increasingly used for the analysis of cohorts of treated PLHIV in sub-Saharan Africa [4, 5]. When this definition is used, active tracing of PLHIV lost to follow-up can help to identify those PLHIV [6, 7] so that strategies to re-engage them in care can be used. In Mali, a resource-limited country without electronic national database, death registries or resources dedicated to active tracing, a definition of LTFU differentiating transient LTFU and definitive LTFU allows to study factors associated with re-engagement in care when no specific programme is implemented to identify and re-engage those PLHIV. It would be useful to identify those among PLHIV lost to follow-up after ART initiation who will subsequently re-engage in care to develop tailored strategies to improve engagement in care of PLHIV who initiate ART.

In Mali, a West African country with 18 million inhabitants, a non-governmental organisation, "Association de Recherche, de Communication et d'Accompagnement à Domicile des Personnes Vivant avec le VIH/SIDA" (ARCAD-SIDA), began to provide healthcare to PLHIV in Bamako (the capital of Mali) in 1998, gradually expanding its activities to several regions after November 2001, as previously described [8]. Healthcare for PLHIV became free after mid-2004, with support from the Global Fund, "Ensemble pour une Solidarité Thérapeutique Hospitalière en Réseau" (ESTHER), a French agency for AIDS care in developing countries, and the Malian government [9, 10]. There were 99,000 adults living with HIV in Mali in 2013, 27,000 of whom were treated [9]. The ESTHER initiative ended in December 2015.

Thus, after more than ten years of scaling-up access to ART in Mali, we aimed to assess predictive factors for re-engagement in care in PLHIV lost to follow-up after ART initiation in clinical centres supported by the ESTHER initiative, focusing on structural barriers, such as care in the capital or in the regions, the care centre level of expertise, and the distance from home to the centre.

## Methods

### Study setting

In Mali, HIV care and ART are provided by community outpatient clinics ($n$ = 14) offering consultations with nurses and/or general practitioners (expertise level I), reference outpatient clinics ($n$ = 54) offering consultations with general practitioners, obstetricians and ophthalmologists, minimal medical tests, including CD4, and simple radiology (expertise level II), or hospitals ($n$ = 9) offering the same care plus specialized consultations and medical tests, including viral load (VL), and CT scans (expertise level III). In addition to hospitals, viral load (VL) determinations are performed in two research institutes in Bamako and are easily available for Bamako outpatient clinics. ART programme is often faced with issues of reagents out of stock and VL device maintenance. The ESTHER initiative [10] focused on reinforcing the clinical skills and capacity of health professionals through peer-to-peer partnerships between French hospitals and their African counterparts and provided an electronic medical records system "Evaluation et Suivi Opérationnel des Programmes d'ESTHER" (ESOPE, Epiconcept, France) for monitoring PLHIV care and the effectiveness of ART, which was progressively implemented at five reference outpatient clinics and two hospitals in Bamako, and at four reference outpatient clinics and five hospitals in five regions (Fig 1), as previously described [8]. The PLHIV included in the ESOPE database accounted for 64% of the 51,000 PLHIV in healthcare for HIV in Mali at the last database update on March 31, 2015 [11].

The data collected in the ESOPE database are socio-demographic (sex, date of pregnancy, date of birth, marital status, last attained educational level, professional activity, sector of employment, residence), clinical (WHO stage, weight, height), HIV diagnosis (date and type of HIV), biological (CD4 count, viral load, blood counts, transaminases), ART (date of the start of ART and type of ART), date of transfer to another care centre and date of death. In ESOPE no clinical diagnosis is collected except presence of tuberculosis, hepatitis B (HBs antigen) or hepatitis C (HCV antibodies) at the start of ART. However, the rate of missing data for these variables and other variables at the start of ART such as viral load, weight and height is more than 80%.

### Study population

This study included all PLHIV who were lost to-follow-up before December 31, 2013, to allow sufficient time to re-engage in care before the last database update (03/31/2015). These PLHIV lost to follow-up were selected from all HIV-1-infected adults, aged $\geq$ 18 years, who began ART between January 1, 2006 and December 31, 2012 at one of the 16 care centres contributing to the ESOPE database, thus beginning treatment at least two years before the last database update (31 March 2015), and who returned for their one-month visit. One month's treatment was initially supplied and the PLHIV were asked to return to the clinical centre on days 15 and 30 to maximise adherence and report any possible adverse events. Treatment was then provided for three months at a time, with clinical visits scheduled every three months and biological tests (CD4 count and VL) performed every six months [9].

At the end of 2005, first national guidelines on HIV care and ART were released, with subsequent revisions in 2008, 2010 and 2013 according to WHO guidelines. In 2005, guidelines recommended treatment for all PLHIV with CD4 counts $\leq$ 200 cells/μL, at WHO stage 3 and CD4 counts $\leq$ 350 cells/μL, or at WHO stage 4. In 2008 and 2010, all PLHIV with CD4 counts $\leq$ 350 cells/μL or at WHO stage 3–4 had to be treated. Pregnant women beginning ART at WHO stage 1–2 and/or CD4 counts > 350 cells/μL were excluded from this study, because ART initiated for pregnancy was stopped after delivery if the clinical stage and

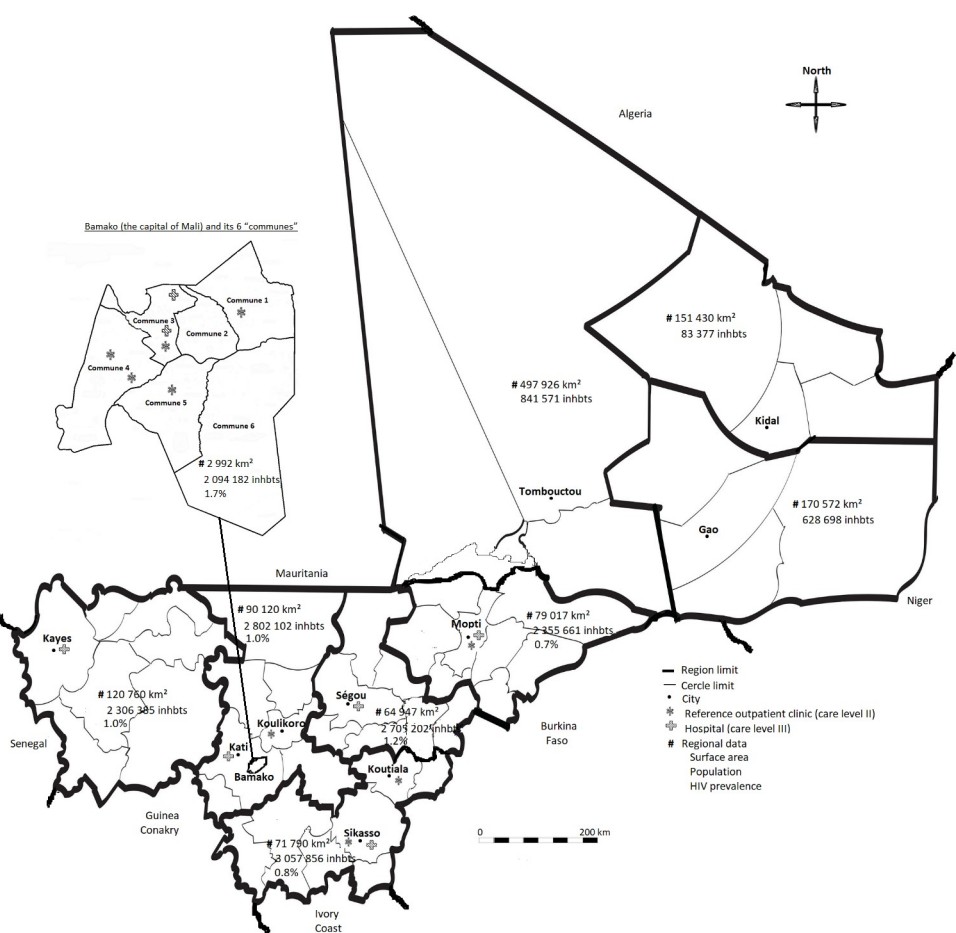

**Fig 1. Map of Mali and its capital (Bamako), showing the location of the study care centres**∗. Abbreviations: Inhbts, inhabitants; km², square kilometres. ∗Data are from the 2013 national demographic and health survey, except for the three northern regions (Gao, Kidal, and Tombouctou), where the survey was not conducted because of the political crisis in 2012 and for which only demographic estimates for 2011 are available. The communes of Bamako are the administrative boroughs of Bamako; circles show the administrative areas in Mali. Overall, in Mali, there are 14 level I, 54 level II, and 9 level III care centres in the regions contributing to the ESOPE database, of which the three regions in northern Mali do not participate (12 additional centres).

biological results at the time of treatment initiation were not in the range of the indication for treatment in the national guidelines for ART [9].

In Mali, the recommended initial ART regimen for HIV-1-infected individuals consists of two nucleoside reverse transcriptase inhibitors (NRTI), including lamivudine or emtricitabine, plus one non-nucleoside reverse transcriptase inhibitor (NNRTI), nevirapine or efavirenz. In 2010, stavudine was removed from the recommended initial ART regimen. The currently recommended initial ART regimen is tenofovir plus lamivudine and efavirenz [9].

## Definitions

Loss to follow-up (LTFU) was defined as an interval of at least six months without any clinical visit for PLHIV not known to be dead or to have been transferred elsewhere. The date of LTFU was the date of the first missed scheduled clinical visit [5, 12], as clinician's first consider an individual to be LTFU at the time of the first missed visit.

We assessed factors associated with subsequent re-engagement in care among people lost to follow-up by defining re-engagement in care as having at least one clinical visit, with or without biological monitoring, in the same ART initiation centre for PLHIV after a period of follow-up interruption of at least six months, as previously defined. The date of re-engagement in care was the date of the first clinical and/or biological visit after LTFU period.

## Statistical analyses

We estimated the cumulative incidence of re-engagement in care and assessed factors associated with this event in PLHIV lost to follow-up. The baseline for this analysis was the date of LTFU. The time without follow-up was counted from the date of LTFU to at the date of re-engagement in care, three years after the date of LTFU or closure of the database (03/31/2015), whichever occurred first. The cumulative incidence of re-engagement in care was estimated using Kaplan-Meier. Factors associated with re-engagement in care were assessed using univariable and multivariable Cox models [13] allowing the comparison of PLHIV who returned to care after LTFU ('return to care' group) and those who did not return to care after LTFU ('extended LTFU' group). The considered individual characteristics at ART initiation were a combined variable of sex and pregnancy, age, WHO stage and CD4 count, period of ART initiation, marital status, last attained educational level, professional activity and a variable combining the location (Bamako or one of the regions), expertise level (outpatient clinic or hospital) of the care centre and its distance from the individual's home. The age at ART initiation was divided into three categories using terciles. The ART initiation periods studied were selected to coincide with the changes in national guidelines. The last attained educational level in ESOPE database was divided into five categories: none, primary, koranic (Franco-Arabic school), secondary and higher which were combined into three categories (none, primary/Koranic, secondary and higher). Professional activity categories were obtained by combining types and sectors of employment, housewife without any professional activity, public or private sector salaried employment, various non-salaried activities, such as being self-employed, farmer/fisherman, other (unemployed including students) and missing. The distance from home to the care centre was determined, in kilometres (km), using Google Maps, by calculating the distance, by road, between the individual's village or neighbourhood of residence and his care centre. In the effort to improve access to healthcare, the Malian government decentralized the outpatient clinics to get them closer (i.e. less than 5 km away) to the majority of the population. According to the third Malian Socio-Sanitary Development Programme (PRODESS III) 2014–2018, the population living less than 5 km from the outpatient clinic rose from 29% in 1998 to 56% in 2012 [14]. We therefore decided to divide the distance from home to care centre into 3 categories: short ($< 5$ km), intermediate ($[5$–$50$ [km) and long ($\geq 50$ km) distance to evaluate the impact of short and long distances on the subsequent re-engagement in care. We were unable to take into account VL at the time of LTFU in our models due to a high rate ($> 80\%$) of missing data. In addition to individual characteristics at ART initiation, we also assessed the association between the subsequent re-engagement in care after LTFU and the following categorical variables: the duration of ART until LTFU ($< 6$, $[6$–$12$ [and $\geq 12$ months) and the estimation of the 12-month change in CD4 count between ART initiation and LTFU ($< 50$ and $\geq 50$ cells/μL). We assessed the interaction between the duration of ART until LTFU and the estimation of the 12-month change in CD4 counts before LTFU because the change in CD4 usually depends on the time on ART. We assessed the interactions between region of care, care centre expertise level and distance from home to care centre because recruitment of PLHIV can be different between the different centres and regions according to distance from home. Given the revisions of national guidelines on HIV care and ART, we also

assessed the interactions between the period of ART initiation and the following variables: Sex and pregnancy, age, WHO stage and CD4 count, and distance from home to care centre. All variables with p-value < 0.05 from Wald test in univariable analyses and covariables used in the multiple imputation (age and marital status) were considered in multivariable analyses. Indeed, the covariables included in the multiple imputation should be used in the final model according to the missing at random data assumption [15]. Statistical significance level was p-value < 0.05. Multicollinearity was checked by calculating the variance inflation factor (VIF).

Among PLHIV lost to follow-up, we estimated the changes in CD4 counts over time by calculating the 12-month change in CD4 counts between ART initiation and LTFU, according to subsequent re-engagement in care or not. The estimation of the 12-month change in CD4 counts before LTFU was obtained by dividing the difference between the CD4 counts at the start of ART and at LTFU by the duration of ART until LTFU in days and multiplying the quotient by 365.25 for each individual. We also estimated the 12-month change in CD4 counts after LTFU for those re-engaged in care. Mann-Whitney test was used to compare the medians of the CD4 counts at ART initiation and of the estimation of the 12-month changes in CD4 counts between groups, and between Bamako and regions. Kruskal-Wallis test was used to compare the medians of the estimation of the 12-month changes in CD4 counts between each type of care centre.

Multiple imputations were performed with chained equations [16, 17] to accommodate missing values, assuming data were conditionally missing at random. More specifically, missing data were imputed for CD4 count, WHO stage, and educational level at the start of ART, and CD4 count at LTFU. We fitted linear regression, logistic regression, and multinomial regression models for CD4 counts at the start of ART and at LTFU (fourth root), WHO stage (1–2 or 3–4), and educational level (none, primary/koranic, secondary and higher), respectively. Multiple imputation is a valid approach for all missing at random mechanisms, whilst complete case analysis may give biased results when the chance of being a complete case depends on the observed values of the outcome (for example LTFU or death) [18]. When data are missing at random, any systematic differences between the observed and missing data can be explained by associations of the missing data with the observed data; for example if CD4 count measurement was more likely to be missing among PLHIV who did not re-engaged in care ('extended LTFU' group) but only because, as they were more likely to be in low socio-economic position, they were less likely to attend the clinical visit where CD4 count was measured [18]. Multiple imputation analyses would avoid bias only if enough variables predictive of missing values are included in the imputation model [15]. Variables at the start of ART accounted for in the imputation models included age, sex, pregnancy, period of ART initiation, region of care and care expertise level, marital status, death (yes or no), and the follow-up time in days ($\log_{10}$) as covariables. The model for CD4 count at LTFU included CD4 count at the start of ART, age at LTFU, sex, period of ART initiation, region of care and care expertise level, and duration of follow-up before LTFU in days ($\log_{10}$). We created 10 and 20 imputed datasets for variables at the start of ART and CD4 count at LTFU, respectively, because of respective percentages of missing data; analyses were run separately on each dataset and the results were combined by Rubin's method [19].

Analyses were conducted using SAS 9.4 (SAS Institute Inc., NC, USA). The study was approved by the national AIDS programme in Mali (approval letter available) in collaboration with the non-governmental organisations ARCAD-SIDA and ESTHER. The study is based on data already collected and does not require new recruitment. Before any medical recording (paper or electronic), it is explained to each PLHIV that data collected can be used to estimate indicators for national AIDS programme evaluation or for clinical research. Written or oral informed consent was obtained from each PLHIV before medical recording.

## Results

Of the 8,037 HIV-positive adults contributing to the ESOPE database and starting ART between 2006 and 2012 with CD4 or WHO stage data available at the start of ART, 7,975 individuals returned for their one-month visit and 4,658 were lost to follow-up before the database closure (03/31/2015) (Fig 2). Among those PLHIV lost to follow-up, 3,650 were lost to follow-up before 12/31/2013 and were included in this study (Fig 2). The study included 2,380 (65%) women and median (interquartile range [IQR]) age was 35 (29–43). The most prescribed initial ART regimen contained zidovudine (AZT) or stavudine (d4T) and lamivudine (3TC) plus nevirapine (NVP) or efavirenz (EFV) for 2,958 (81%) individuals. The second most prescribed one contained tenofovir (TDF) and emtricitabine (FTC) or 3TC plus NVP or EFV for 429 (12%) individuals. Overall median duration of ART before LTFU was 11 months (IQR, 5–22) (S1 Table).

### Incidence of re-engagement in care

Among the 3,650 PLHIV lost to follow-up before 2013, 1,975 (54%) were definitively LTFU during the subsequent three-year follow-up period ('extended LTFU' group) and 1,675 (46%) subsequently returned to care within the same time interval ('return to care' group). The Kaplan-Meier cumulative incidence rate of re-engagement in care was 39.0% (95%CI, 38.0–41.0) at one year after the date of LTFU, 45.0% (95%CI, 43.0–47.0) at two years, and 47.0% (95%CI, 45.0–48.0) at three years.

### Factors associated with re-engagement in care

All variance inflation factors were less than 2, so there was no multicollinearity. We found an interaction between the region of care and care expertise level and the distance from home to the care centre, and we combined these three variables. We also found an interaction between duration of ART until LTFU and the estimation of the 12-month change in CD4 count, and we combined these two variables.

   Factors associated with re-engagement in care in the univariable and multivariable analyses are described in Table 1. At the start of ART, pregnant women (aHR 1.56; 95%CI 1.30–1.86 *versus* males), people at WHO stage 1–2 and CD4 counts ≥ 200 cells/μL (aHR 1.23; 95%CI 1.07–1.41 versus people at WHO stage 3–4 and CD4 counts < 200 cells/μL), and those who started ART in 2008–2009 (aHR 1.20; 95%CI 1.06–1.36 versus 2006–2007) had a higher likelihood of re-engagement in care. Conversely, PLHIV who started ART in 2010–2012 (aHR 0.56; 95%CI 0.49–0.64) were less likely to re-engage in care than those initiating ART in 2006–2007 (Table 1). Marital status and professional activity were not associated with re-engagement in care, except farmers/fishermen who were at lower risk of re-engagement in care compared to housewives.

   Re-engagement in care rates did not differ according to the distance between home and the care centre among PLHIV followed at Bamako outpatient clinics (Table 2). However, PLHIV followed at Bamako hospitals [5–50 [km from their homes (aHR 0.91; 95%CI 0.79–1.04), regional outpatient clinics > 5 km from their homes (aHR 0.78; 95%CI 0.56–1.09 for [5–50 [km and aHR 0.64; 95%CI 0.41–1.01 for ≥ 50 km), and regional hospitals (aHR 0.36; 95%CI 0.28–0.47 for [5–50 [km and aHR 0.33; 95%CI 0.21–0.51 for ≥ 50 km), were less likely to re-engage in care than PLHIV followed at Bamako outpatient clinics (Table 1). In addition, in multivariable analysis comparing Bamako and regions within each care centre level of expertise (outpatient clinics or hospitals), PLHIV followed in Bamako were more likely to re-engage in care than those followed in the regions (aHR 2.20; 95%CI 1.71–2.84 for hospitals and aHR 1.21; 95%CI 0.99–1.49 for outpatient clinics) (S2 Table).

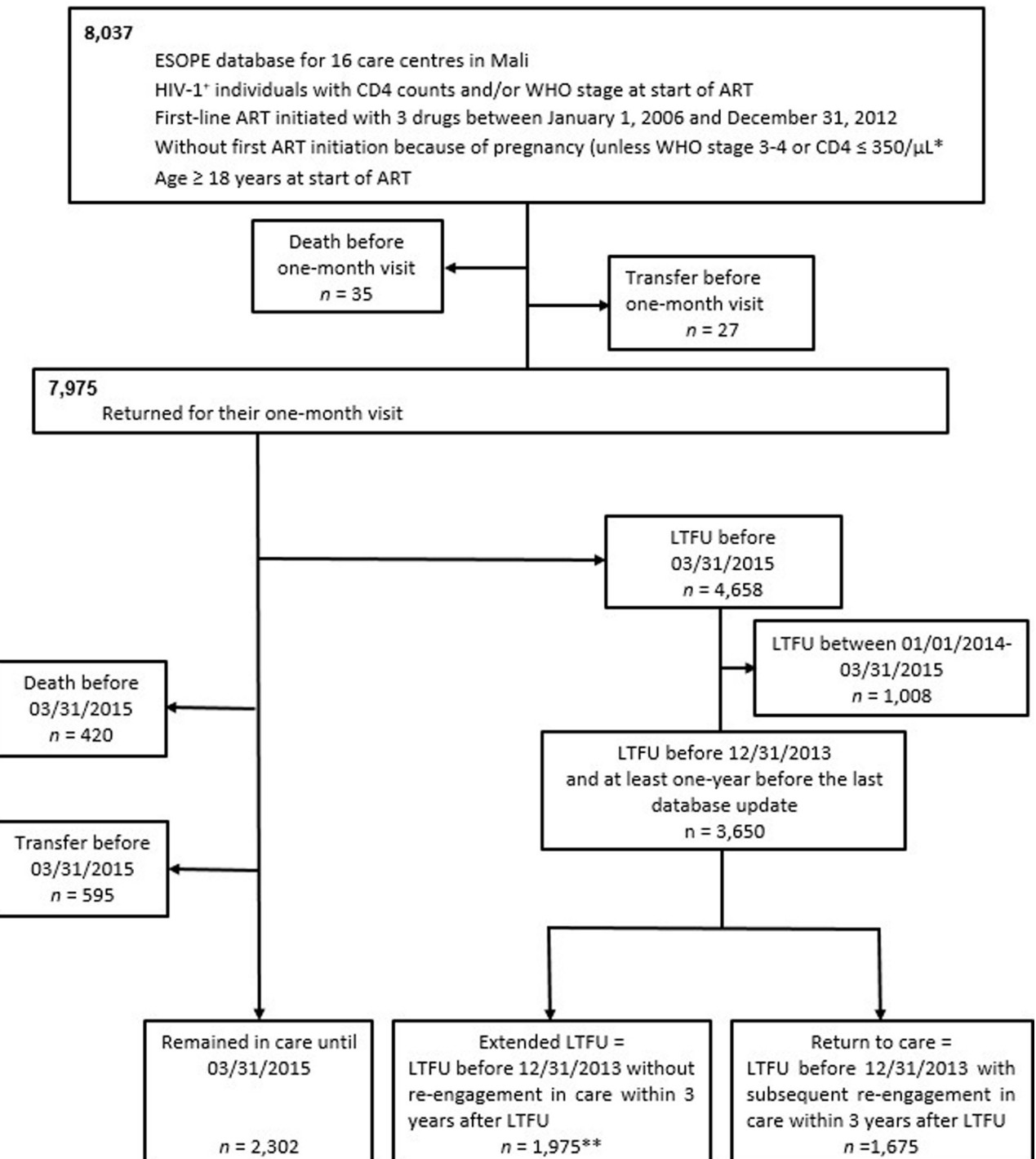

**Fig 2. Flowchart of group selection.** *Pregnant women beginning ART at WHO stage 1–2 and/or with a CD4 count > 350 cells/μL were excluded from this study because ART initiated for pregnancy was stopped after delivery if the clinical stage and biological results at the time of treatment initiation were not within the range of the indication for treatment in the national guidelines for ART. ** Only 27 PLVIH re-engaged in care after 3 years of LTFU.

Furthermore, PLHIV with a duration of ART until LTFU of [6–12 [months with an estimated 12-month change in CD4 count < 50 cells/μL (aHR 1.46; 95%CI 1.23–1.73) or an

**Table 1. Factors associated with re-engagement in care in the 36 months after LTFU among people living with HIV starting ART in Mali subsequent to LTFU (Cox model with imputed data).**

| Characteristics | Extended LTFU N = 1,975 | Return to care N = 1,675 | Univariable | | Multivariable | |
|---|---|---|---|---|---|---|
| | n (%) | n (%) | HR (95% CI) | P | HR (95% CI) | P |
| **Sex and pregnancy** | | | | < 0.0001 | | < 0.0001 |
| Men | 723 (36.6) | 547 (32.7) | 1 | | 1 | |
| Non-pregnant women | 1,171 (59.3) | 926 (55.3) | 1.04 (0.94–1.15) | | 1.04 (0.91–1.19) | |
| Pregnant women | 81 (4.1) | 202 (12.0) | 2.10 (1.81–2.43) | | 1.56 (1.30–1.86) | |
| **Age (years)** | | | | 0.1980 | | 0.1502 |
| < 30 | 551 (27.9) | 485 (29.0) | 1 | | 1 | |
| [30–40] | 761 (38.5) | 673 (40.2) | 1.02 (0.91–1.14) | | 1.10 (0.98–1.24) | |
| ≥ 40 | 663 (33.6) | 517 (30.8) | 0.92 (0.82–1.04) | | 1.04 (0.91–1.19) | |
| **WHO stage and CD4 count (cells/μL)** (missing WHO stage, 9%; missing CD4, 14%) | | | | < 0.0001 | | 0.0102 |
| WHO stage 3–4 and CD4 < 200 | 662 (33.5) | 404 (24.1) | 1 | | 1 | |
| WHO stage 3–4 and CD4 ≥ 200 | 274 (13.9) | 208 (12.4) | 1.19 (0.99–1.43) | | 1.08 (0.90–1.30) | |
| WHO stage 1–2 and CD4 < 200 | 612 (31.0) | 573 (34.2) | 1.33 (1.17–1.52) | | 1.12 (0.98–1.28) | |
| WHO stage 1–2 and CD4 ≥ 200 | 427 (21.6) | 490 (29.3) | 1.54 (1.35–1.75) | | 1.23 (1.07–1.41) | |
| **Periods of ART initiation** | | | | < 0.0001 | | < 0.0001 |
| 2006–2007 | 598 (30.3) | 817 (48.8) | 1 | | 1 | |
| 2008–2009 | 391 (19.8) | 479 (28.6) | 0.96 (0.87–1.07) | | 1.20 (1.06–1.36) | |
| 2010–2012 | 986 (49.9) | 379 (22.6) | 0.41 (0.36–0.46) | | 0.56 (0.49–0.64) | |
| **Marital status** | | | | <0.0001 | | 0.0094 |
| Monogamous[a] | 869 (44.0) | 699 (41.7) | 1 | | 1 | |
| Polygamous | 358 (18.1) | 353 (21.1) | 1.14 (1.01–1.29) | | 1.06 (0.94–1.20) | |
| Single | 251 (12.7) | 226 (13.5) | 1.09 (0.94–1.25) | | 0.99 (0.85–1.15) | |
| Widowed | 221 (11.2) | 240 (14.3) | 1.22 (1.06–1.40) | | 1.08 (0.94–1.25) | |
| Divorced | 97 (4.9) | 97 (5.8) | 1.17 (0.95–1.43) | | 1.04 (0.85–1.28) | |
| Missing | 179 (9.1) | 60 (3.6) | 0.50 (0.38–0.64) | | 0.57 (0.43–0.77) | |
| **Educational level** (missing, 15%) | | | | 0.0283 | | 0.1779 |
| None | 994 (50.3) | 759 (45.3) | 1 | | 1 | |
| Primary/Koranic | 578 (29.2) | 525 (31.3) | 1.13 (1.00–1.28) | | 1.03 (0.91–1.17) | |
| Secondary | 289 (14.7) | 286 (17.1) | 1.19 (1.03–1.38) | | 1.12 (0.97–1.30) | |
| Higher | 114 (5.8) | 105 (6.3) | 1.13 (0.90–1.42) | | 1.04 (0.81–1.34) | |

(*Continued*)

**Table 1.** (Continued)

| Characteristics | Extended LTFU N = 1,975 | Return to care N = 1,675 | Univariable | | Multivariable | |
|---|---|---|---|---|---|---|
| | n (%) | n (%) | HR (95% CI) | P | HR (95% CI) | P |
| **Professional activity[b]** | | | | <0.0001 | | 0.1477 |
| Housewife | 1,435 (39.3) | 768 (38.9) | 1 | | 1 | |
| Public sector staff | 209 (5.7) | 104 (5.3) | 1.09 (0.90–1.33) | | 1.02 (0.82–1.27) | |
| Private sector staff | 691 (18.9) | 341 (17.3) | 1.11 (0.98–1.25) | | 1.04 (0.90–1.20) | |
| Self-employed | 521 (14.3) | 287 (14.5) | 0.96 (0.84–1.11) | | 1.09 (0.92–1.29) | |
| Farmer/Fisherman | 243 (6.7) | 164 (8.3) | 0.63 (0.50–0.79) | | 0.78 (0.61–1.00) | |
| Other | 433 (11.9) | 223 (11.3) | 1.06 (0.91–1.22) | | 0.98 (0.83–1.16) | |
| Missing | 118 (3.2) | 88 (4.4) | 0.49 (0.34–0.70) | | 1.17 (0.81–1.16) | |
| **Regions, care centre & distance (km)** | | | | < 0.0001 | | < 0.0001 |
| Bamako outpatient clinics | 834 (42.2) | 1,005 (60.0) | 1 | | 1 | |
| Bamako hospitals & < 5 | 51 (2.6) | 68 (4.0) | 1.08 (0.86–1.36) | | 1.22 (0.97–1.54) | |
| Bamako hospitals & [5–50[ | 349 (17.7) | 300 (17.9) | 0.82 (0.73–0.93) | | 0.91 (0.79–1.04) | |
| Bamako hospitals & ≥ 50 | 52 (2.6) | 55 (3.3) | 0.97 (0.74–1.26) | | 1.18 (0.92–1.52) | |
| Regional outpatient clinics & < 5 | 95 (4.8) | 96 (5.7) | 0.93 (0.76–1.14) | | 0.96 (0.74–1.23) | |
| Regional outpatient clinics & [5–50[ | 74 (3.8) | 38 (2.3) | 0.53 (0.39–0.72) | | 0.78 (0.56–1.09) | |
| Regional outpatient clinics & ≥ 50 | 37 (1.9) | 21 (1.3) | 0.58 (0.38–0.88) | | 0.64 (0.41–1.01) | |
| Regional hospitals & [5–50[ | 355 (17.9) | 71 (4.2) | 0.25 (0.19–0.31) | | 0.36 (0.28–0.47) | |
| Regional hospitals & ≥ 50 | 128 (6.5) | 21 (1.3) | 0.20 (0.13–0.31) | | 0.33 (0.21–0.51) | |
| **Duration of ART until LTFU (month) and 12-month change in CD4 count (cell/µL)[c] (missing CD4 at LTFU, 44%)** | | | | < 0.0001 | | < 0.0001 |
| < 6 | 711 (36.0) | 338 (20.2) | 1 | | 1 | |
| [6–12 [and < 50 | 283 (14.3) | 227 (13.6) | 1.50 (1.26–1.77) | | 1.46 (1.23–1.73) | |
| [6–12 [and ≥ 50 | 101 (5.1) | 270 (16.1) | 3.15 (2.66–3.72) | | 2.61 (2.21–3.08) | |
| ≥ 12 and < 50 | 641 (32.5) | 250 (14.9) | 0.90 (0.75–1.08) | | 0.87 (0.72–1.04) | |
| ≥ 12 and ≥ 50 | 239 (12.1) | 590 (35.2) | 3.09 (2.70–3.53) | | 2.43 (2.12–2.78) | |

Abbreviations: LTFU, loss to follow-up; ART, antiretroviral therapy; HR, hazards ratio; CI, confidence interval.

[a]Married or living with a partner.

[b]Professional activities: professional activity categories combined types and sectors of employment, housewife without any professional activity, public or private sector salaried employment, various non-salaried activities, such as being self-employed, farmer/fisherman, and other (unemployed including students).

[c]Variable combining the duration of ART until LTFU (month) and the estimation of the 12-month change in CD4 count (cells/ µL).

**Table 2. Multivariable Hazards Ratio (HRs) for re-engagement in the 36 months after loss to follow-up (LTFU) in each type of care centre according to the distances (km) from home (Cox model with imputed data).**

| Distance to care centre (km)* | Extended LTFU | Return to care | Multivariable | |
|---|---|---|---|---|
| | | | HR (95% CI) | *P* |
| **Bamako outpatient clinics** | N = 834 | N = 1,005 | | *0.6196* |
| | n (%) | n (%) | | |
| < 5 | 194 (23.3) | 215 (21.4) | 1 | |
| [5–50[ | 593 (71.1) | 742 (73.8) | 0.96 (0.83–1.11) | |
| ≥ 50 | 47 (5.6) | 48 (4.8) | 0.91 (0.65–1.26) | |
| **Bamako hospitals** | N = 452 | N = 423 | | *0.0299* |
| | n (%) | n (%) | | |
| < 5 | 51 (11.3) | 68 (16.1) | 1 | |
| [5–50[ | 349 (77.2) | 300 (70.9) | 0.76 (0.60–0.97) | |
| ≥ 50 | 52 (11.5) | 55 (13.0) | 0.91 (0.66–1.25) | |
| **Regional outpatient clinics** | N = 206 | N = 155 | | *0.0798* |
| | n (%) | n (%) | | |
| < 5 | 95 (46.1) | 96 (61.9) | 1 | |
| [5–50[ | 74 (35.9) | 38 (24.5) | 0.73 (0.48–1.10) | |
| ≥ 50 | 37 (18.0) | 21 (13.6) | 0.63 (0.38–1.06) | |
| **Regional hospitals** | N = 483 | N = 92 | | *0.8114* |
| | n (%) | n (%) | | |
| [5–50[ | 355 (73.5) | 71 (77.2) | 1 | |
| ≥ 50 | 128 (26.5) | 21 (22.8) | 1.07 (0.63–1.82) | |

* Adjusted on sex & pregnancy, age, WHO stage and CD4 count, period of ART initiation, marital status, education level, professional activity and a combined variable of the duration of ART until LTFU (months) and the estimation of the 12-month change in CD4 count (cells/μL).

estimated 12-month change in CD4 count ≥ 50 cells/μL (aHR 2.61; 95%CI 2.21–3.08), and those with a duration of ART until LTFU ≥ 12 months with an estimated 12-month change in CD4 count ≥ 50 cells/μL (aHR 2.43; 95%CI 2.12–2.78) were more likely to re-engage in care than those with a duration of ART until LTFU < 6 months regardless the estimation of the 12-month change in CD4 count. PLHIV with a duration of ART until LTFU ≥ 12 months with an estimated 12-month change in CD4 count < 50 cells/μL were less likely to re-engage in care (Table 1).

## Changes in CD4 count according to region and expertise level of the care centre

Table 3 shows the median (IQR) of CD4 count at ART initiation, LTFU, and return to care; duration of ART until LTFU; duration of LTFU; and 12-month change in CD4 count in each group ('return to care' or 'extended LTFU') according to region and care expertise level. The CD4 count at LTFU was missing for 1,606 (44%) PLHIV. Median time between ART initiation and LTFU was 10 months (IQR, 5–25) for the 'extended LTFU' group and 12 months (IQR, 7–21) for the 'return to care' group (Table 3). Median CD4 count at ART initiation (137 *vs.* 163 cells/μL) and median 12-month change in CD4 count after ART initiation (0 vs. 64 cells/μL) were lower in the 'extended LTFU' group than in the 'return to care' group (p < 0.0001 for both) (Table 3). In the 'return to care' group, median 12-month change in CD4 count was higher in Bamako than in the regions, either before LTFU (71 cells/μL; IQR, 0–177 and 20 cells/μL; IQR, 0–133, respectively) or after LTFU (75 cells/μL; IQR, -165-396 and 31 cells/μL; IQR, -150-356, respectively) (p < 0.0001). In the 'extended LTFU' group, the 12-month change

**Table 3. Median (IQR) CD4 count at ART initiation, LTFU, and return to care; duration of ART until LTFU; duration of LTFU; and 12-month change in CD4 count in each group (return to care or extended LTFU) according to region and care centre expertise level (imputed data).**

| | CD4 count at ART initiation (cells/μL) | CD4 count at LTFU (cells/μL) | Duration of ART until LTFU (months) | 12-month change in CD4 count (cells/μL)* | p-value** | CD4 count at return to care (cells/μL) | Duration of LTFU (months) | 12-month change in CD4 count (cells/μL)*** | p-value** |
|---|---|---|---|---|---|---|---|---|---|
| | | | | Return to care n = 1,675 | | | | | |
| **Region** | | | | | | | | | |
| Bamako (n = 1,428) | 166 (66–278) | 290 (162–459) | 12 (8–21) | 71 (0–177) | < 0.0001 | 346 (193–528) | 5 (4–9) | 75 (-165-396) | < 0.0001 |
| Regions (n = 247) | 144 (55–258) | 226 (94–368) | 11 (6–20) | 20 (0–133) | | 242 (110–451) | 5 (3–10) | 31 (-165-396) | |
| **Region and care expertise level** | | | | | | | | | |
| Bamako outpatient clinics (n = 1,005) | 167 (64–277) | 288 (157–454) | 14 (8–23) | 63 (0–164) | < 0.0001 | 339 (185–517) | 5 (4–9) | 68 (-170-387) | < 0.0001 |
| Bamako hospitals (n = 423) | 163 (68–283) | 303 (177–482) | 10 (7–16) | 95 (0–234) | | 368 (215–543) | 4 (3–9) | 96 (-144-429) | |
| Regional outpatient clinics (n = 155) | 149 (56–260) | 236 (100–364) | 10 (6–18) | 14 (0–112) | | 236 (114–460) | 5 (3–10) | 25 (-150-384) | |
| Regional hospitals (n = 92) | 136 (48–239) | 222 (87–381) | 14 (6–22) | 30 (0–149) | | 244 (101–430) | 6 (4–10) | 35 (-151-304) | |
| Overall (n = 1,675) | 163 (64–273) | 283 (152–447) | 12 (7–21) | 64 (0–173) | - | 331 (177–518) | 5 (4–9) | 69 (-162-388) | - |
| | | | | Extended LTFU n = 1,975 | | | | | |
| **Region** | | | | | | | | | |
| Bamako (n = 1,286) | 136 (47–255) | 159 (55–291) | 10 (5–23) | 0 (0–28) | - | | | | - |
| Regions (n = 689) | 139 (50–258) | 147 (52–292) | 12 (5–28) | 0 (-19-25) | | | | | |
| **Region and care expertise level** | | | | | | | | | |
| Bamako outpatient clinics (n = 834) | 134 (45–249) | 155 (52–278) | 8 (4–21) | 0 (0–20) | - | - | - | - | - |
| Bamako hospitals (n = 452) | 141 (49–267) | 167 (64–320) | 12 (5–26) | 0 (0–41) | | - | - | - | |
| Regional outpatient clinics (n = 206) | 158 (64–288) | 214 (95–375) | 11 (5–24) | 0 (-1-75) | | - | - | - | |
| Regional hospitals (n = 483) | 132 (44–245) | 127 (44–254) | 12 (5–31) | 0 (-25-11) | | - | - | - | |
| Overall (n = 1,975) | 137 (48–257) | 156 (54–291) | 10 (5–25) | 0 (-2-27) | - | - | - | - | - |

* 12-month change in CD4 cell count before LTFU = [(CD4 at LTFU—CD4 at ART initiation)/(date of LTFU-date of ART initiation)] x 365.25

** p-values for comparison of 12-month change in CD4 count between Bamako and regions, and between each type of care centre were from Mann-Whitney and Kruskal-Wallis test, respectively

*** 12-month change in CD4 cell count after LTFU = [(CD4 return date—CD4 LTFU)/ (return date—date of LTFU)] x 365.25

in CD4 count before LTFU was null for the different types of care centre (Table 3). Among the 3,650 PLHIV lost to follow-up before 2013, only 301 (8%) and 536 (15%) had VL at the start of ART and at the time of LTFU, respectively. Nearly 95% of PLHIV who received VL measurements were from Bamako.

## Discussion

Among individuals lost to follow-up, the cumulative incidence rates of subsequent re-engagement in the care centre where ART was initiated rose from 39.0% (95% CI: 38.0–41.0) at one year to 47.0% (95% CI: 45.0–48.0) at three years. Pregnant women, people with good immunological status (WHO stage 1–2 and CD4 counts $\geq$ 200 cells/μL) at the start of ART and those starting ART during earlier periods (2006–2009) were more likely to re-engage in care. People followed at Bamako hospitals [5–50 [km from their home, regional hospitals or regional outpatient clinics $\geq$ 5 km from their homes were less likely to re-engage in care than those followed at Bamako outpatient clinics. People followed at Bamako were more likely to re-engage in care than those followed in the regions, regardless of care centre expertise level. PLHIV treated for at least 6 months regardless of a 12-month CD4 count gain between ART initiation and LTFU, and those treated for more than 12 months with a 12-month CD4 count gain above 50 cells/μL were more likely to re-engage in care than those treated for less than 6 months. However, PLHIV treated for more than 12 months with a 12-month CD4 count gain below 50 cells/μL between ART initiation and LTFU were less likely to re-engage in care.

In our study, the cumulative incidence rates of re-engagement in care were high, particularly in the first year after LTFU. We only ascertained re-engagement in care when PLHIV re-engaged in the care centre from which they were lost to follow-up, and could not thus estimate the overall re-engagement in care which may be higher. In South Africa, Ambia et al matched individual clinical data with the data from the demographic and health surveillance system and found a cumulative incidence of re-engagement in the same centre or in another centre of 23.0% (95% CI: 20.3–25.8) and 38.1% (95% CI: 33.1–43.0) at one and three years after LTFU, respectively [20]. These estimates are slightly lower than ours. In contrast, a multicentric study of 3 countries (Uganda, Kenya and Tanzania) in East Africa [21] that traced randomly selected PLHIV lost to follow-up observed a cumulative incidence of re-engagement in care at one year after LTFU of 13.3% (95% CI: 11.1–15.3) versus 10.0% (95% CI: 9.1–10.8) in those not traced. These estimates were significantly lower than ours. We excluded from our analyses PLHIV who were known to have been transferred to another care centre or to have died. However, in the absence of tracing, the cumulative incidence in our study may be overestimated due to the failure to account for the competing risks of silent transfers to another centre and unknown deaths among PLHIV lost to follow-up [22]. Indeed, a meta-analysis of individual data from studies in sub-Saharan Africa that traced PLHIV lost to follow-up estimated high cumulative incidences of silent transfer and death among PLHIV lost to follow-up, 11.9% (95% CI: 11.1–12.6) and 14.6% (95% CI: 13.8–15, 4) at one and three years after LTFU respectively for silent transfer, and 20.2% (95% CI: 19.3–21.1) and 21.8% (95% CI: 20.8–22.7) at one and three years after LTFU respectively for death [23]. Also, they observed that the cumulative incidences of silent transfers and death were the highest in the first year after LTFU [23]. Furthermore, some of PLHIV could be misclassified as lost to follow-up if some of their clinical visits are not well recorded in the database [7, 24], the cumulative incidence rate of re-engagement in care in the first year after LTFU could also be overestimated.

Active tracing of PLHIV [25] and matching to death registries [26] allows better estimations of the outcomes of individuals LTFU, i.e. death, silent transfer to other clinical centres, or disengagement from care. The Malian national ART programme has no dedicated resources for

the active tracing of PLHIV and there is no national death registry. Although it is well known that PLHIV lost to follow-up are at higher risk of death due to treatment interruption, we were unable to determine whether PLHIV lost to follow-up had died [27–29]. A higher risk of mortality was found in PLHIV with severe immune depression (CD4 counts < 200 cells/μL) [8,30]. Also, we were unable to investigate silent transfers because some PLHIV classified as LTFU may have been silent transfers to another care centre [7, 24].

The main strength of this study was the large number of PLHIV, accounting for 64% under care in Mali, followed either in the capital or in five other regions. Centres participating in this study were those with the two highest levels of expertise and seven of nine hospitals in the territory covered by the study participated in the database.

With the significant growth of the Malian ART programme since 2010 [11] and the availability of clinics nearer to individuals' homes [31], silent transfer could explain the lower rate of re-engagement in care of PLHIV who started ART in 2010–2012. People with an advanced clinical stage or a CD4 count below 200 cells/μL at the start of ART had a lower rate of re-engagement in care. The absence of a rapid clinical response to treatment in the most immunosuppressed people may result in poor adherence to ART [32, 33]. It is also possible that some of these people died, as suggested by others [7, 23]. People treated for more than 12 months with an estimated 12-month change in CD4 count before LTFU of less than 50 cells/μL had a lower rate of re-engagement in care which could be explained by poor adherence. Poorer adherence to ART before LTFU, resulting in a smaller gain in CD4 count, could explain an extended LTFU without subsequent re-engagement in care [32, 33]. In contrast, people treated for more than 12 months with an estimated 12-month change in CD4 count before LTFU above than 50 cells/μL, who had a higher rate of re-engagement in care, may have had a well-controlled infection due to a better adherence to ART allowing increase in CD4 count. Furthermore, it is well known that men access care later than women, with an advanced clinical stage and low CD4 count [34, 35].

PLHIV followed in Bamako were more likely to re-engage in care than those followed in the regions and the gain of CD4 counts before LTFU was higher in Bamako than elsewhere. Access to viral load monitoring is higher in Bamako than in regions. However, the achievement of viral load remains low due to frequent and repeated ruptures of reagents and consumables, and maintenance problems with viral load devices. Access to viral load monitoring may improve the monitoring of treatment efficacy and readjustment of treatment when necessary. A lower time to viral suppression and better decrease of viral load on ART has been associated with early retention in care [36]. The gain of CD4 counts after LTFU, higher in Bamako than in the regions, could indicate that PLHIV actually remained on ART. Explanations could be non-adherence to the national guidelines, with the provision of ART for more than three months at a time [9], poor filling out of the medical record, with some PLHIV incorrectly labelled as LTFU [37], or silent transfers. In East African studies, people who reported barriers related to poverty (including transportation costs) and poor quality of care (including roughness of the staff, disrespect, scolding, and/or too much time spent at the care centre) were more likely to re-engage in care should these conditions change or when feeling ill [7, 38,39]. In our study, being followed in regional hospital or regional outpatient clinic ≥ 5 km away was associated with a lower rate of re-engagement in care, which could be related to transportation issues [7].

Several factors were reported by PLHIV lost to follow-up as barriers to subsequent re-engagement in care, such as perceived stigma [6, 7], problems related to poverty or access to care centre (lack of food, lack of money, difficult access and expensive transportation, and too long waiting time at the care centre), problems related to the medical staff (disrespect, discrimination and scolding) [7, 37–39]. Also, PLHIV who felt well and did not feel the need for care

were less likely to re-engage in care [7, 38, 40]. We did not have the possibility to assess those data, or to assess social support from the family and the community, and HIV status disclosure. We also did not show differences according to social data such as educational level contrary to what has been shown in studies of LTFU risk factors [8, 41]. However, difficulties to return to care when living more than 5 km to the care centre in the regions could be due to financial or transportation issues.

In conclusion, our study identified several individual and structural factors associated with less re-engagement in care such as advanced clinical stage and low CD4 count, starting ART in recent period, being followed in regional hospitals or regional outpatient clinics ≥ 5 km from home to care centre, being treated less than 6 months, and being treated more than 12 months with low gain in CD4 count during follow-up. Furthermore, according to UNAIDS data, 32% of PLHIV in Mali are currently on ART and 41% of them show viral suppression [42]. Thus, to achieve the UNAIDS 90-90-90 targets, the Malian ART programme may need to invest in comprehensive intervention packages throughout the country that increase earlier screening of PLHIV before advanced clinical stage, ART uptake and retention in care, such as enhancing ART adherence with community-based adherence support intervention, decentralized provision of ART, and early peer tracing strategy or others tracing strategies. A Continuing the care decentralization programme already launched could help PLHIV experiencing difficulties of engagement in care due to transportation issues to respect the scheduled clinical visits. More, implementing access to viral load monitoring throughout the country, an electronic national database and an electronic death registry seems necessary to achieve the best quality of care by the way of adapted political decisions.

## Supporting information

**S1 Table. Characteristics at the start of antiretroviral therapy (ART), duration of ART until LTFU and the estimated 12-month change in CD4 count between ART initiation and LTFU of individuals lost to follow-up (LTFU) before 12/31/2013, overall and by group ("extended LTFU" and "return to care") (data not imputed).**
(DOCX)

**S2 Table. Multivariable Hazards Ratio (HRs) for re-engagement in the 36 months after loss to follow-up (LTFU) in each expertise level of care centre according to the region (Cox model with imputed data).**
(DOCX)

## Acknowledgments

We would like to thank all the participants of the Malian ART programme and research assistants from Mali and France.

Aliou BALDE is supported by a doctoral contract from Sorbonne Université (ex Université Pierre et Marie-Curie, UPMC, Paris 6). All authors declare no conflict of interest, although some authors have, at some stage in the past, received funding from a variety of pharmaceutical companies for research, travel grants, speaking engagements or consultancy fees.

This research did not receive any specific grant from funding agencies in the public, commercial, or not-for-profit sectors.

## Author Contributions

**Conceptualization:** Aliou Baldé, Dominique Costagliola, Sophie Abgrall.

**Data curation:** Aliou Baldé, Dominique Costagliola, Sophie Abgrall.

**Formal analysis:** Aliou Baldé.

**Methodology:** Dominique Costagliola, Sophie Abgrall.

**Supervision:** Dominique Costagliola, Sophie Abgrall.

**Validation:** Dominique Costagliola, Sophie Abgrall.

**Writing – original draft:** Aliou Baldé.

**Writing – review & editing:** Aliou Baldé, Laurence Lièvre, Almoustapha Issiaka Maiga, Fodié Diallo, Issouf Alassane Maiga, Dominique Costagliola, Sophie Abgrall.

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
