## [Decision Letter · Decision Letter 0]

17 Jan 2020

PONE-D-19-19274

Re-engagement in care of people living with HIV lost to follow-up after initiation of antiretroviral therapy in Mali: who returns to care?

PLOS ONE

Dear Mr. BALDE,

Thank you for submitting your manuscript to PLOS ONE. After careful consideration, we feel that it has merit but does not fully meet PLOS ONE’s publication criteria as it currently stands. Therefore, we invite you to submit a revised version of the manuscript that addresses the points raised during the review process.

We would appreciate receiving your revised manuscript by Mar 01 2020 11:59PM. To enhance the reproducibility of your results, we recommend that if applicable you deposit your laboratory protocols in protocols.io, where a protocol can be assigned its own identifier (DOI) such that it can be cited independently in the future. For instructions see: http://journals.plos.org/plosone/s/submission-guidelines#loc-laboratory-protocols

We look forward to receiving your revised manuscript.

Kind regards,

Carmen Melatti

Academic Editor PLOS ONE

On behalf of 

Joseph Fokam, Ph.D

Academic Editor

PLOS ONE

Reviewers' comments:

Reviewer's Responses to Questions

**Comments to the Author**

1. Is the manuscript technically sound, and do the data support the conclusions?

Reviewer #1: Yes

Reviewer #2: No

Reviewer #3: No

2. Has the statistical analysis been performed appropriately and rigorously? 

Reviewer #1: I Don't Know

Reviewer #2: No

Reviewer #3: Yes

3. Have the authors made all data underlying the findings in their manuscript fully available?

Reviewer #1: Yes

Reviewer #2: No

Reviewer #3: No

4. Is the manuscript presented in an intelligible fashion and written in standard English?

Reviewer #1: Yes

Reviewer #2: No

Reviewer #3: Yes

5. Review Comments to the Author

Reviewer #1: The study described by Aliou Balde and collaborators aimed at assessing risk factors for first loss-to-follow-up (LFTU) and subsequent re-engagement in the original HIV care for PLHIV in Mali. LTFU remains one of the biggest challenge for attainment of the UNAIDS 90-90-90 HIV treatment target. The paper reads well and is written in a simple language. Nevertheless, the paper needs some clarifications to improve the quality.

Below are suggestions for authors to consider:

- The study period (2006-2012) is a bit older and falling in a period where WHO recommendations were quite different of current guidelines towards the 90-90-90 UNAIDS initiative, just wondering if findings of this study are not obsolete; in addition, why excluding participants who were LTFU after December 2013, if extended re-engagement evaluation was for 3 years and we are now in 2019: will suggest inclusion of more recent data; so that corrective measures could have a more positive impact in HIV follow up and long term sustainability of current ART programs.

- The difference between people followed at Bamako hospitals and Bamako outpatient clinics is not clear to the reader, please be more precise.

- Change in CD4 counts over time was estimated using a formula not supported by a reference, please indicate the reference. In addition, was it possible for the authors (based on medical history/record) to identify co-infection such as TB that could have an impact on CD4 variation during LTFU?

- Authors did not provide the change in plasmatic viral load which is the most reliable parameter to evaluate ART program outcome: Will suggest the use of available viral load data (though representing a small group when compared to the overall sample size) to estimate the impact of LTFU on viral load; Authors mentioned that viral load measurement was available for 301 and 536 study subjects at the start of ART and the time of LTFU, respectively.

- Authors failed to indicate number of transfer (595) and deaths (420) in the result section as shown in figure2, a bit confusing for the reader since the sum of LTFU and non-LTFU does not give the number enrolled. This information can be inserted in the paragraph on “factors associated with re-engagement in care”.

- Evaluation of factors associated with LTFU such as stigma could strengthen the study and help decision makers to improve the health system.

- Typos? CD4 count is usually given in cells/mm3 or cells/µl but not /µl only.

Reviewer #2: Review report

In this paper, the authors provide interesting data about re-engagement in care of PLHIV after a period of LTFU. However, the study has several limitations and the statistical analysis has to be improved to make the conclusion more reliable. Furthermore, Authors have published results on LTFU in 2019: and they described LTFU in their papers: The current paper might be concentrated in re-engagement as primary outcome instead of repeating data on LTFU in the same population:

https://pubmed.ncbi.nlm.nih.gov/30270487-risk-factors-for-loss-to-follow-up-transfer-or-death-among-people-living-with-hiv-on-their-first-antiretroviral-therapy-regimen-in-mali/?from_term=Risk+factors+for+loss+to+follow+up%2C+transfer+or+death+among+people+living+on+their+first+antiretroviral+therapy+regimen+in++Mali&from_pos=1

The manuscript should more clearly compare the re-engagement incidence estimates of PLHIV in care of this study with other African studies and studies from other continents.

The Authors should note that PLOS One publication criteria state that if a submitted study replicates or is very similar to previous work, authors must provide a sound scientific rationale for the submitted work and clearly reference and discuss the existing literature. Submissions that replicate or are derivative of existing work will likely be rejected if authors do not provide adequate justification:

(https://journals.plos.org/plosone/s/criteria-for-publication#loc-2)

Method:-How did they measure the socio-economic status? We did not found such data in the manuscript. They have reported Professional activity: Did they use that as socio-economic status?

Method: How education is categorized and defined needs to be added in the method section or as footnote. When Authors are reporting secondary: is this the attained or the achieved level?

To analyze the Re-engagement issue: How authors did measure the duration from LTFU and re-engagement: What was the end date since this is the outcome.

Method:-: The information on 12-month change in CD4 count, does not make much sense. Did they a strong evidence to use this way to calculate this variable?

Method:- Authors should report the analytic approaches they used in the method section: which statistics tests, what was the significance alpha value . Furthermore, in my opinion it would be important to indicate Which confounders were used in the adjusted models? Authors should define clearly the threshold for significance (alpha).

For step-wise multiple regression analyses: Authors should report the alpha level used; discuss whether the variables were assessed for collinearity and interaction; describe the variable selection process by which the final model was developed (e.g., forward-stepwise; ect…).

Since Authors compare median , they should , detail any post hoc tests that were performed.

Manuscripts submitted to PLOS ONE are expected to report statistical methods in sufficient detail for others to replicate the analysis performed. Ensure that results are rigorously reported in accordance with community standards and that the statistical methods employed are appropriate for the study design.

- Results, Table1 and 3 must be rewritten : It would be better to mention the exact p-values for each category instead of reporting overall p-value. for example: when the authors reported HR: 1.03(0.95-1.11) and used p-value of 0.0013 for 1-2 and CD4≥200: this will confuse reader. Tables 1 and 3 must be totally rewritten.

- Results, Why did the authors categorize age and did not use it as continuous variable? Furthermore, the authors should revise the categories of variables : Age and Regions, care and distance: There are overlapping between: 30-40 and ≥40. Same thing with distance

- Results: he authors have introduced sex and professional activity in their models: Did they check for multicollinearity? Did they check the VIF?

Results: The authors mention several information (data) that were not mentioned in the any tables or figures: They should provide Supporting information( “S” and number. For example, “S1 Appendix” and “S2 Appendix,” “S1 Table” and “S2 Table,” and so forth) to support the statements provided in this manuscript.

- Discussion: The discussion needs to be rewritten. The authors mention several information (data) that were not mentioned in the results and the comparison to other studies is insufficient.

- Conclusion in the abstract is reporting ART adherence and transportation issues which were not collected and analyzed in this study. Conclusions must be presented in an appropriate fashion and must be supported by the data

- References must be rewritten according to PLOS One policies: PLOS uses the numbered citation (citation-sequence) method and first six authors, et al.

- Authors are citing unpublished work in the manuscript ( LTFU rates did not differ according to the distance between home and the care centre among the PLHIV followed at Bamako outpatient clinics( data not shown)):

- According to the Plos One policies: Do not cite the following sources in the reference list: Unavailable and unpublished work, including manuscripts that have been submitted but not yet accepted (e.g., “unpublished work,” “data not shown”). Instead, include those data as supplementary material or deposit the data in a publicly available database.

Conclusion: MAJOR REVISION

Reviewer #3: In this study, Baldé and colleagues assess the risk factors for patient LTFU and subsequent re-engagement in HIV care at the clinic in which they enrolled in Mali. Strengths of the study include its large sample size and sophisticated analyses. However, overall enthusiasm for the study is weakened by time period of the data (2006-2013), which may not be as relevant today in the era of universal ART eligibility and the broader scale-up of HIV care that has occurred over time, and that the findings do not add particularly novel data to the current LTFU literature. Moreover, the lack of data on undocumented deaths and silent transfers (which the authors acknowledge) is a major limitation in understanding the factors associated with re-engagement in care for those who become lost, particularly given that >50% of the total study population became lost to follow-up using the authors’ definition.

Abstract:

- There is no mention in the methods about using multiply imputation

Introduction:

- The authors write “It would also useful to identify those among PLHIV LTFU after ART initiation who will subsequently re-engage in care to develop tailored strategies to improve linkage to care of PLHIV who initiate ART.” However, the recommendations given by the authors in the discussion / conclusion are only general policy recommendations. How do the authors believe that the findings in this study can specifically be used to address their statement in the introduction and have an impact on the current HIV epidemic?

- “…jeopardize treatment efficacy and illness outcome, particularly by raising the risk of the acquisition of viral resistance mutations…” this statement could use a citation

Methods:

- The authors write “First, we assessed factors associated with LTFU by defining LTFU as an interval of at least six months without any clinical appointment for PLHIV not known to be dead or to have been transferred elsewhere.” Followed by “The date of LTFU was the date of the last centre visit plus three months (i.e. the date of the missed scheduled clinical visit) [4, 13].” These two sentences seem to contain contradictory definitions - the first LTFU definition is 6 months and the second is 3 months. The references cited do not help clarify this either, as the first reference (4) gives a summary of LTFU definitions and the second (13) used a definition of “loss to follow-up defined as a duration of >6 months between the last visit recorded and the closing date of the database”

- Why were 5 km and 50 km used cutoffs for distance from the clinic?

- Time on ART is an important variable on care outcomes and is not included in the models

Results:

- The tables contain densities of information and overwhelm the reader

- The % missing for each of the variables would be better listed in the tables or text as they were difficult to locate in the footnotes of such large tables

- Presuming the tables show the results of the imputation analysis, there are no complete case analysis results shown or any comparison between the complete-case and imputation results

- Table 1, WHO stage and CD4 count (cell/μL) row, Not LTFU column - does not sum to 100%

Discussion:

- The first two paragraphs of the discussion recap results that have already been presented. It would be more informative to the reader to understand what the authors’ believe to be the significance of their findings.

- What would the authors believe to be cause of the observed variability in likelihood of re-engagement across the different time periods?

- Imputation is risky if the imputed variables are not missing at random. Why do the authors believe that having a CD4 count, for example, is MAR and not also related to some of the same social-ecological factors that drive retention in care?

Other general comments:

- The writing could use some grammatical editing. For example “LTFU” is defined as “loss to follow-up” however the phrase “PLHIV LTFU” is used repeatedly, and stating “people living with hiv loss-to-follow-up” doesn’t make sense grammatically.

6. PLOS authors have the option to publish the peer review history of their article (what does this mean?). If published, this will include your full peer review and any attached files.

Reviewer #1: Yes: Aubin Nanfack

Reviewer #2: Yes: Akilimali Zalagile Pierre

Reviewer #3: No

---

## [Author Response · Author response to Decision Letter 0]

5 Mar 2020

Letter to the Editor and Reviewers

February 29, 2020

Dear Editor and Reviewers,

Thank you for the opportunity to revise and resubmit our manuscript.

Enclosed, please find the responses to each point raised by the reviewers (see below), and the revised manuscript with and without track changes.

We decided to focus on re-engagement in care and we removed factors associated with LTFU (previous table 1) from the manuscript. Also, professional activity variable has been removed from our Cox models. We therefore modified the manuscript accordingly. All the data used in the manuscript can be found in the tables. Additional tables were added in supplementary files.

Sincerely yours,

Aliou BALDE

 

Response to reviewers

Review Comments to the Author

Reviewer #1: The study described by Aliou Balde and collaborators aimed at assessing risk factors for first loss-to-follow-up (LFTU) and subsequent re-engagement in the original HIV care for PLHIV in Mali. LTFU remains one of the biggest challenge for attainment of the UNAIDS 90-90-90 HIV treatment target. The paper reads well and is written in a simple language. Nevertheless, the paper needs some clarifications to improve the quality.

Below are suggestions for authors to consider:

- The study period (2006-2012) is a bit older and falling in a period where WHO recommendations were quite different of current guidelines towards the 90-90-90 UNAIDS initiative, just wondering if findings of this study are not obsolete; in addition, why excluding participants who were LTFU after December 2013, if extended re-engagement evaluation was for 3 years and we are now in 2019: will suggest inclusion of more recent data; so that corrective measures could have a more positive impact in HIV follow up and long term sustainability of current ART programs.

Answer: Thank you for this comment. We did not have data for periods after 2015 because the “Ensemble pour une Solidarité Thérapeutique Hospitalière en Réseau” (ESTHER) initiative ended in December 2015. ESTHER, a French agency for AIDS in developing countries, provided since 2005 an electronic medical records system “Evaluation et Suivi Opérationnel des Programmes d’ESTHER” (ESOPE, Epiconcept, France) in order to monitor PLHIV care and effectiveness of ART. This system was gradually implemented in 16 care centres in Bamako and regions with a financial support from ESTHER. After ESTHER funding ended in December 2015, the global fund, the main funder of the Malian ART programme did not renew funding for the ESOPE database. We therefore do not have updated data after December 2015. We think that in a country which depends on external funds for the HIV care, it is important to show through our study the necessity of continuing the implementation of an electronic data system throughout the country for a better assessment of the quality of care and of the continuum of care.

- The difference between people followed at Bamako hospitals and Bamako outpatient clinics is not clear to the reader, please be more precise.

Answer: We now provide an additional table in supplementary files (supplementary table 2) which describe our study population (n=7,975), overall and by region and care centre expertise level. Nevertheless, we performed the Chi2 or Kruskal-Wallis test for the comparison of the characteristics of individuals between each type of care centre and we found that all the p-values were significant (<0.0001) (p-values not shown in Supplementary table 2).

The following sentence was added in the results section: 

 “Characteristics at the start of ART of the individuals were slightly different according to the region of care and the care centre expertise level.”

- Change in CD4 counts over time was estimated using a formula not supported by a reference, please indicate the reference. In addition, was it possible for the authors (based on medical history/record) to identify co-infection such as TB that could have an impact on CD4 variation during LTFU?

 Answer: For the 12-month change in CD4 count we just estimated the variation, gain or loss between two CD4 counts (at start of ART and at LTFU) per unit of time (year). We do not understand why we need to support it by a reference. 

We agree that TB co-infection could have an impact on CD4 variation. In ESOPE, no clinical diagnosis is collected except TB at the start of ART. However, the rate of missing data for this variable is above 80%. We cannot take into account TB co-infection either at the start of ART or during follow-up. Therefore, we cannot account for this comment. 

For more details on data collected in ESOPE, we added the following sentences in the study setting section: 

“The data collected in the ESOPE database are socio-demographic (sex, date of pregnancy, date of birth, marital status, last attained educational level, professional activity, sector of employment, residence), clinical (WHO stage, weight, height), HIV diagnosis (date and type of HIV), biological (CD4 count, viral load, blood counts, transaminases), ART (date of the start of ART and type of ART), date of transfer to another care centre and date of death. In ESOPE no clinical diagnosis is collected except presence of tuberculosis, hepatitis B (HBs antigen) or hepatitis C (HCV antibodies) at the start of ART. However, the rate of missing data for these variables and other variables at the start of ART such as viral load, weight and height is more than 80%.”

- Authors did not provide the change in plasmatic viral load which is the most reliable parameter to evaluate ART program outcome: Will suggest the use of available viral load data (though representing a small group when compared to the overall sample size) to estimate the impact of LTFU on viral load; Authors mentioned that viral load measurement was available for 301 and 536 study subjects at the start of ART and the time of LTFU, respectively. 

 Answer: We agree that the viral load is the most reliable parameter to evaluate ART programme outcome. Malian ART programme received its first viral load devices in 2011. Efforts are being made to improve access to viral load but regular viral load measurements were not sufficiently implemented over our study period. Among the 3,650 PLHIV lost to follow-up before 2013 there were only 913 (25%) PLHIV who benefited from a viral load measurement. So, viral load measurement was available for 301 (8%) PLHIV at the start of ART, 536 (15%) PLHIV at the time of LTFU and 401 (11%) PLHIV at the time of re-engagement in care. We did not impute variables with more than 80% of missing data such as viral load. So, we cannot take this comment into account.

- Authors failed to indicate number of transfer (595) and deaths (420) in the result section as shown in figure 2, a bit confusing for the reader since the sum of LTFU and non-LTFU does not give the number enrolled. This information can be inserted in the paragraph on “factors associated with re-engagement in care”.

 Answer: We decided to remove the previous Table 1 on “Risk factors for being LTFU among people living with HIV starting ART in Mali (Competing risk Cox model accounting for known transfers and deaths)” from the manuscript and to focus on re-engagement in care. 

Nevertheless, we added in supplementary files (supplementary table 1) a description of the characteristics of individuals including missing data by follow-up status (Remained in care, Loss to follow-up (LTFU) before 12/31/2013, LTFU after 12/31/2013, known Transfer to another centre and known Death) at the start of antiretroviral therapy (ART).

We also changed the first two paragraphs of the result section as follows:

“Of the 8,307 HIV-positive adults contributing to the ESOPE database and starting ART between 2006 and 2012 with CD4 or WHO stage data available at the start of ART, 7,975 individuals returned for their one-month visit and were included in this study (Fig. 2). The individual characteristics including missing data are described in Supplementary Table 1 based on their subsequent follow-up status (‘Remained in care’, ‘LTFU before 12/31/2013’, ‘LTFU after 12/31/2013’, ‘Transfer’ or ‘Death’). The percentages of missing data were 14% for CD4 cell count, 9% for WHO stage, and 15% for educational level. During the study period, 2,302 (29%) PLHIV remained in care, 3,650 (46%) and 1,008 (12%) were lost to follow-up for at least six months during their follow-up before and after 12/31/2013, respectively. Also, during the study period, 595 (8%) and 420 (5%) were known to be transferred to another care centre and to be dead, respectively (S1 Table). 

In addition, Supplementary Table 2 (S2 Table) shows the characteristics of individuals, overall and according to the region of care and the care centre expertise level at the start of ART (imputed data). The study included 5,330 (67%) women and median (interquartile range [IQR]) age was 36 (29-43). The most prescribed initial ART regimen contained zidovudine (AZT) or stavudine (d4T) and lamivudine (3TC) plus nevirapine (NVP) or efavirenz (EFV) for 6,106 (77%) individuals. The second most prescribed one contained tenofovir (TDF) and emtricitabine (FTC) or 3TC plus NVP or EFV for 1,332 (17%) individuals. Overall median follow-up was 26 months (IQR, 8-48) (S2 Table).”

- Evaluation of factors associated with LTFU such as stigma could strengthen the study and help decision makers to improve the health system.

 Answer: We decided to focus on re-engagement in care and we removed factors associated with LTFU from the manuscript. Also, we did not collect data on stigma in this study. Nevertheless, we agree with this comment and we think that stigma solving particularly in regions may improve the retention in care as suggested in our previous study (Baldé A et al, HIV Med 2019). 

- Typos? CD4 count is usually given in cells/mm3 or cells/µl but not /µl only.

Answer: We agree with this comment and we corrected the typos of CD4 count by using cells/µL.

 

Reviewer #2: Review report

In this paper, the authors provide interesting data about re-engagement in care of PLHIV after a period of LTFU. However, the study has several limitations and the statistical analysis has to be improved to make the conclusion more reliable. Furthermore, Authors have published results on LTFU in 2019: and they described LTFU in their papers: The current paper might be concentrated in re-engagement as primary outcome instead of repeating data on LTFU in the same population:

https://pubmed.ncbi.nlm.nih.gov/30270487-risk-factors-for-loss-to-follow-up-transfer-or-death-among-people-living-with-hiv-on-their-first-antiretroviral-therapy-regimen-in-mali/?from_term=Risk+factors+for+loss+to+follow+up%2C+transfer+or+death+among+people+living+on+their+first+antiretroviral+therapy+regimen+in++Mali&from_pos=1

 Answer: Thank you for this comment. We agree with this comment. We decided to focus on re-engagement in care and to remove factors associated with LTFU from the manuscript. 

The manuscript should more clearly compare the re-engagement incidence estimates of PLHIV in care of this study with other African studies and studies from other continents.

The Authors should note that PLOS One publication criteria state that if a submitted study replicates or is very similar to previous work, authors must provide a sound scientific rationale for the submitted work and clearly reference and discuss the existing literature. Submissions that replicate or are derivative of existing work will likely be rejected if authors do not provide adequate justification:

(https://journals.plos.org/plosone/s/criteria-for-publication#loc-2)

Answer: We agree with this comment. We now compared our cumulative incidence rates of re-engagement in care to other sub-Saharan Africa studies results. 

The following sentences were added in the discussion section: 

“In our study, the cumulative incidence rates of re-engagement in care were high, particularly in the first year after LTFU. We only ascertained re-engagement in care when PLHIV re-engaged in the care centre from which they were lost to follow-up, and could not thus estimate the overall re-engagement in care which may be higher. In South Africa, Ambia et al matched individual clinical data with the data from the demographic and health surveillance system and found a cumulative incidence of re-engagement in the same centre or in another centre of 23.0% (95% CI: 20.3-25.8) and 38.1% (95% CI: 33.1-43.0) at one and three years after LTFU, respectively [20]. These estimates are slightly lower than ours. In contrast, a multicentric study of 3 countries (Uganda, Kenya and Tanzania) in East Africa [21] that traced randomly selected PLHIV lost to follow-up observed a cumulative incidence of re-engagement in care at one year after LTFU of 13.3% (95% CI: 11.1-15.3) versus 10.0% (95% CI: 9.1-10.8) in those not traced. These estimates were significantly lower than ours. We excluded from our analyses PLHIV who were known to have been transferred to another care centre or to have died. However, in the absence of tracing, the cumulative incidence in our study may be overestimated due to the failure to account for the competing risks of silent transfers to another centre and unknown deaths among PLHIV lost to follow-up [22]. Indeed, a meta-analysis of individual data from studies in sub-Saharan Africa that traced PLHIV lost to follow-up estimated high cumulative incidences of silent transfer and death among PLHIV lost to follow-up, 11.9% (95% CI: 11.1-12.6) and 14.6% (95% CI: 13.8-15, 4) at one and three years after LTFU respectively for silent transfer, and 20.2% (95% CI: 19.3-21.1) and 21.8% (95% CI: 20.8-22.7) at one and three years after LTFU respectively for death [23]. Also, they observed that the cumulative incidences of silent transfers and death were the highest in the first year after LTFU [23]. Furthermore, some of PLHIV could be misclassified as lost to follow-up if some of their clinical visits are not well recorded in the database [7, 24], the cumulative incidence rate of re-engagement in care in the first year after LTFU could also be overestimated.” 

Method:-How did they measure the socio-economic status? We did not found such data in the manuscript. They have reported Professional activity: Did they use that as socio-economic status?

Method: How education is categorized and defined needs to be added in the method section or as footnote. When Authors are reporting secondary: is this the attained or the achieved level?

To analyze the Re-engagement issue: How authors did measure the duration from LTFU and re-engagement: What was the end date since this is the outcome.

Answer: We now describe data collected in ESOPE in the study setting section. We do not have income data in the ESOPE database and we used the occupation (Professional activity) to measure the socioeconomic status. The more conventional indicators of socioeconomic status were completed schooling and occupation (Duncan GJ et al, Am J Public Health 2002; Mackenbach JP et al, N Engl J of Med 2008). Nevertheless, education is the most stable measure of socioeconomic position, because it is normally completed early in adulthood, avoiding most of the problems of reverse causation (Mackenbach JP et al, Lancet Public Health 2019). The educational level collected in ESOPE database is the “The last attained educational level”. Following your comment, the professional activity variable has now been removed from our Cox models. Therefore, we removed all the socioeconomic terms in our manuscript.

We added the following sentences in the statistical analyses section: 

“The last attained educational level in ESOPE database was divided into five categories: none, primary, koranic (Franco-Arabic school), secondary and higher which were combined into three categories (none, primary/Koranic, secondary and higher).” 

 

Method:-: The information on 12-month change in CD4 count, does not make much sense. Did they a strong evidence to use this way to calculate this variable?

 Answer: For the 12-month change in CD4 count we just estimated the gain between two CD4 counts (at start of ART and at LTFU) per unit of time (year). There were only 301 (8%) PLHIV at the start of ART and 536 (15%) PLHIV at the time of LTFU who benefited from a viral load measurement. With the non-availability of viral load, we believe that the gain of CD4 cells before the interruption of care (Loss to follow-up) may make sense to differentiate PLHIV whose virus might be controlled by the ART allowing increase in CD4 count. However, we performed a multivariable analysis without this variable and found that the hazards ratios for the other risk factors did not change (see table at below). 

 

Table. Factors associated with re-engagement in care in the 36 months after LTFU among people living with HIV starting ART in Mali subsequent to LTFU (multivariable Cox model without combined variable of duration on ART before LFTU (month) and 12-month change in CD4 count (cells/µL))

Characteristics at start of ART Extended LTFU

N = 1,975 Return to care

N = 1,675 

Multivariable

 n (%) n (%) HR (95% CI) P 

Sex and pregnancy

Men

Non-pregnant women

 Pregnant women 

723 (36.6)

1,171 (59.3)

81 (4.1) 

547 (32.7)

926 (55.3)

202 (12.0) 

1

1.09 (0.97-1.22)

1.79 (1.52-2.10) < 0.0001

Age (years)

< 30

[30-40[

≥ 40 

551 (27.9)

761 (38.5)

663 (33.6) 

485 (29.0)

673 (40.2)

517 (30.8) 

1

1.12 (1.00-1.31)

1.08 (0.95-1.23) 0.1281

WHO stage and CD4 count (cells/µL) 

WHO stage 3-4 and CD4 < 200

WHO stage 3-4 and CD4 ≥ 200

WHO stage 1-2 and CD4 < 200

WHO stage 1-2 and CD4 ≥ 200 

662 (33.5)

274 (13.9)

612 (31.0)

427 (21.6) 

404 (24.1)

208 (12.4)

573 (34.2)

490 (29.3) 

1

1.14 (0.95-1.36)

1.14 (0.99-1.31)

1.30 (1.13-1.49) 0.0013

Periods of ART initiation

2006-2007

2008-2009

2010-2012 

598 (30.3)

391 (19.8)

986 (49.9) 

817 (48.8)

479 (28.6)

379 (22.6) 

1

1.16 (1.04-1.30)

0.50 (0.44-0.57) < 0.0001

Marital status

Monogamousa 

Polygamous

Otherb 

869 (44.0)

358 (18.1)

748 (37.9) 

699 (41.7)

353 (21.1)

623 (37.2) 

1

1.05 (0.93-1.19)

1.01 (0.91-1.13) 0.6029

Educational level

None

Primary/Koranic

Secondary

Higher 

994 (50.3)

578 (29.2)

289 (14.7)

114 (5.8) 

759 (45.3)

525 (31.3)

286 (17.1)

105 (6.3) 

1

1.07 (0.94-1.21)

1.14 (0.99-1.31)

1.10 (0.87-1.39) 0.1372

Regions, care centre & distance (km) 

Bamako outpatient clinics 

Bamako hospitals & < 5

Bamako hospitals & [5-50[

Bamako hospitals & ≥ 50

Regional outpatient clinics & < 5

Regional outpatient clinics & [5-50[

Regional outpatient clinics & ≥ 50

Regional hospitals & [5-50[

Regional hospitals & ≥ 50 

834 (42.2)

51 (2.6)

349 (17.7)

52 (2.6)

95 (4.8)

74 (3.8)

37 (1.9)

355 (17.9)

128 (6.5) 

1,005 (60.0)

68 (4.0)

300 (17.9)

55 (3.3)

96 (5.7)

38 (2.3)

21 (1.3)

71 (4.2)

21 (1.3) 

1

1.19 (0.95-1.49)

0.83 (0.73-0.95)

1.08 (0.82-1.41)

0.94 (0.75-1.18)

0.79 (0.58-1.08)

0.60 (0.38-0.92)

0.29 (0.22-0.36)

0.25 (0.17-0.38) < 0.0001

Abbreviations: LTFU, loss to follow-up; ART, antiretroviral therapy; HR, hazards ratio; CI, confidence interval. aMarried or living with a partner. bOther marital status: single, divorced, widowed, or missing (6%). 

 

Method:- Authors should report the analytic approaches they used in the method section: which statistics tests, what was the significance alpha value . Furthermore, in my opinion it would be important to indicate Which confounders were used in the adjusted models? Authors should define clearly the threshold for significance (alpha).

 Answer: Now, all the analytic approaches used in our study were provided in the statistical analyses section. Also, we defined the threshold for significance (alpha). We did not look for confounding variables. 

For step-wise multiple regression analyses: Authors should report the alpha level used; discuss whether the variables were assessed for collinearity and interaction; describe the variable selection process by which the final model was developed (e.g., forward-stepwise; ect…).

Since Authors compare median, they should, detail any post hoc tests that were performed.

Manuscripts submitted to PLOS ONE are expected to report statistical methods in sufficient detail for others to replicate the analysis performed. Ensure that results are rigorously reported in accordance with community standards and that the statistical methods employed are appropriate for the study design. 

 Answer: We did not perform backward elimination, forward or stepwise selection. However, all variables with p-value < 0.05 from Wald test in univariable analyses and covariables used in the multiple imputation (age and marital status) were considered in multivariable analyses and maintained in the final model as suggested by Sterne et al (Sterne et al, BMJ 2009). Also, the interactions assessed were provided in the statistical analyses section.

The statistical analyses section of our manuscript has been rewritten as follows: 

“We assessed factors associated with re-engagement in care in PLHIV lost to follow-up. For this analysis, PLHIV who were lost to follow-up after December 31, 2013 were excluded to allow sufficient time to re-engage in care before the last database update (03/31/2015). The baseline for this analysis was the date of LTFU. Follow-up time was censored at closure of the database or three years after the date of LTFU, whichever occurred first. Factors associated with re-engagement in care were assessed using univariable and multivariable Cox models [13] allowing the comparison of PLHIV who returned to care after LTFU (‘return to care’ group) and those who did not return to care after LTFU (‘extended LTFU’ group). The considered individual characteristics at ART initiation were a combined variable of sex and pregnancy, age, WHO stage and CD4 count, period of ART initiation, marital status, last attained educational level and a variable combining the location (Bamako or one of the regions), expertise level (outpatient clinic or hospital) of the care centre and its distance from the individual’s home. The ART initiation periods studied were selected to coincide with the changes in national guidelines. The last attained educational level in ESOPE database was divided into five categories: none, primary, koranic (Franco-Arabic school), secondary and higher which were combined into three categories (none, primary/Koranic, secondary and higher). The distance from home to the care centre was determined, in kilometres (km), using Google Maps, by calculating the distance, by road, between the individual’s village or neighbourhood of residence and his care centre. In the effort to improve access to healthcare, the Malian government decentralized the outpatient clinics to get them closer (i.e. less than 5 km away) to the majority of the population. According to the third Malian Socio-Sanitary Development Programme (PRODESS III) 2014-2018, the population living less than 5 km from the outpatient clinic rose from 29% in 1998 to 56% in 2012 [14]. We therefore decided to divide the distance from home to care centre into 3 categories: short (< 5 km), intermediate ([5-50[ km) and long (≥ 50 km) distance to evaluate the impact of short and long distances on the subsequent re-engagement in care. We were unable to take into account VL at the time of LTFU in our models due to a high rate (> 80%) of missing data. In addition to individual characteristics at ART initiation, we also assessed the association between the subsequent re-engagement in care after LTFU and the following categorical variables: the duration of ART until LTFU (< 6, [6-12[ and ≥ 12 months) and the estimation of the 12-month change in CD4 count between ART initiation and LTFU (< 50 and ≥ 50 cells/µL). We assessed the interaction between the duration of ART until LTFU and the estimation of the 12-month change in CD4 counts before LTFU, and the interactions between region of care, care centre expertise level and distance from home to care centre. We also assessed the interactions between the period of ART initiation and the following variables: Sex and pregnancy, age, WHO stage and CD4 count, and distance from home to care centre. All variables with p-value < 0.05 from Wald test in univariable analyses and covariables used in the multiple imputation (age and marital status) were considered in multivariable analyses. Indeed, the covariables included in the multiple imputation should be used in the final model according to the missing at random data assumption [15]. Statistical significance level was p-value < 0.05. Multicollinearity was checked by calculating the variance inflation factor (VIF).

Among PLHIV lost to follow-up, we estimated the changes in CD4 counts over time by calculating the 12-month change in CD4 counts between ART initiation and LTFU, according to subsequent re-engagement in care or not. The estimation of the 12-month change in CD4 counts before LTFU was obtained by dividing the difference between the CD4 count at the start of ART and at LTFU by the duration of ART until LTFU in days and multiplying the quotient by 365.25 for each individual. We also estimated the 12-month change in CD4 counts after LTFU for those re-engaged in care. Mann-Whitney test was used to compare the medians of the CD4 counts at ART initiation and of the estimation of the 12-month changes in CD4 counts between groups, and between Bamako and regions. Kruskal-Wallis test was used to compare the medians of the estimation of the 12-month changes in CD4 counts between each type of care centre.”

In the sous-section of results on Factors associated with re-engagement in care we added the following sentences:

“All variance inflation factors were less than 2, so there was no multicollinearity. We found an interaction between the region of care and care expertise level and the distance from home to the care centre, and we combined these three variables. We also found an interaction between duration of ART until LTFU and the estimation of the 12-month change in CD4 count, and we combined these two variables.”

- Results, Table1 and 3 must be rewritten: It would be better to mention the exact p-values for each category instead of reporting overall p-value. for example: when the authors reported HR: 1.03(0.95-1.11) and used p-value of 0.0013 for 1-2 and CD4≥200: this will confuse reader. Tables 1 and 3 must be totally rewritten.

Answer: We do not agree with this comment. Even if the variable has several categories, what matters is the p-value of the association of the variable with the outcome, not the p-values of the categories of the variable. The hazards ratio (HR) and 95% confidence interval of the categories of the variable that contain 1 are simply not different from the reference category which have a HR=1. 

Nevertheless, the p-values in univariable analysis of age, marital status and education level were incorrect, we corrected them.

- Results, Why did the authors categorize age and did not use it as continuous variable? Furthermore, the authors should revise the categories of variables: Age and Regions, care and distance: There are overlapping between: 30-40 and ≥40. Same thing with distance

Answer: The effect of a continuous is not necessarily linear. Using 3 categories of age allows to detect presence of a non-linear effect without using more complicated modeling such as spline. In addition, for the distance from home to care centre the 5 km distance correspond to the decentralization policy which was set up in the country. 

We revised the categories of age and distance. The categories for age and distance were presented in the manuscript as follows: < 30 years, [30-40[ years and ≥ 40 years, and < 5 km, [5-50[ km and ≥ 50 km for age and distance, respectively. 

- Results: the authors have introduced sex and professional activity in their models: Did they check for multicollinearity? Did they check the VIF?

Answer: As explain above we removed the professional activity variable from all of our models. In addition, we checked for all of variables the multicollinearity according to re-engagement in care (see results below). We did not find any multicollinearity, our highest Variance Inflation Factor (VIF) was 1.30 (< 5). However, we decided to remove professional activity variable from all of our Cox models. 

Variable Tolerance Variance

Inflation

Intercept . 0

Sex and pregnancy 0.76760 1.30276

Age (years) 0.84030 1.19005

WHO stage and CD4 count (cells/µL) 0.94047 1.06329

Periods of ART initiation 0.88543 1.12940

Marital status 0.96561 1.03562

Education level 0.93471 1.06985

Professional activity 0.88584 1.12888

Regions, care centre & distance (km) 0.92122 1.08551

Duration of ART until LTFU (month) and 12-month change in CD4 count (cell/µL) 0.94663 1.05638

Results: The authors mention several information (data) that were not mentioned in the any tables or figures: They should provide Supporting information ( “S” and number. For example, “S1 Appendix” and “S2 Appendix,” “S1 Table” and “S2 Table,” and so forth) to support the statements provided in this manuscript.

Answer: We now provide in our manuscript the Table 2 which shows the multivariable Hazards Ratio (HRs) for re-engagement in the 36 months after loss to follow-up (LTFU) in each type of care centre according to the distances (km) from home (Cox model with imputed data). 

We added in Table 3 (previous Table 2) the median (IQR) CD4 count at D0, LTFU, and return to care; duration of ART before LTFU; duration of LTFU; and 12-month change in CD4 count in each group (return to care or extended LTFU) according to Bamako and regions. 

Also, we now provide three supplementary tables as follows: 

Supplementary table 1. Characteristics of individuals by follow-up status (Remained in care, Loss to follow-up (LTFU) before 12/31/2013, LTFU after 12/31/2013, known Transfer to another centre and known Death) at the start of antiretroviral therapy (ART) (data not imputed).

Supplementary table 2. Characteristics of individuals, overall and by regions and care centre at the start of antiretroviral therapy (ART) (imputed data).

Supplementary table 3. Multivariable Hazards Ratio (HRs) for re-engagement in the 36 months after loss to follow-up (LTFU) in each expertise level of care centre according to the region (Cox model with imputed data).

- Discussion: The discussion needs to be rewritten. The authors mention several information (data) that were not mentioned in the results and the comparison to other studies is insufficient.

- Conclusion in the abstract is reporting ART adherence and transportation issues which were not collected and analyzed in this study. Conclusions must be presented in an appropriate fashion and must be supported by the data

Answer: We rewrote the discussion and conclusion sections of the manuscript and also the conclusion in the abstract. 

As the factors associated with re-engagement in care were generally reported by PLHIV lost to follow-up in some others studies, we added the following sentences in our discussion section: 

“Several factors were reported by PLHIV lost to follow-up as barriers to subsequent re-engagement in care , such as perceived stigma [6-7], problems related to poverty or access to care centre (lack of food, lack of money, difficult access and expensive transportation, and too long waiting time at the care centre), problems related to the medical staff (disrespect, discrimination and scolding) [7, 37-39]. Also, PLHIV who felt well and did not feel the need for care were less likely to re-engage in care [7, 38, 40]. We did not have the possibility to assess those data, or to assess social support from the family and the community, and HIV status disclosure. We also did not show differences according to social data such as educational level contrary to what has been shown in studies of LTFU risk factors [8 (Baldé et al, HIV Med 2019, 41 (Akilimali et al, Plos one 2017)]. However, difficulties to return to care when living more than 5 km to the care centre in the regions could be due to financial or transportation issues.”

- References must be rewritten according to PLOS One policies: PLOS uses the numbered citation (citation-sequence) method and first six authors, et al.

Answer: We rewrote all the references according to PLOS One policies.

- Authors are citing unpublished work in the manuscript (LTFU rates did not differ according to the distance between home and the care centre among the PLHIV followed at Bamako outpatient clinics (data not shown)):

- According to the Plos One policies: Do not cite the following sources in the reference list: Unavailable and unpublished work, including manuscripts that have been submitted but not yet accepted (e.g., “unpublished work,” “data not shown”). Instead, include those data as supplementary material or deposit the data in a publicly available database.

Conclusion: MAJOR REVISION

 Answer: We now provided in the manuscript the Table 2 showing that re-engagement in care rates did not differ according to the distance between home and the care centre among the PLHIV followed at Bamako outpatient clinics. Also, we provided three additional tables in supplementary files.

 

Reviewer #3: In this study, Baldé and colleagues assess the risk factors for patient LTFU and subsequent re-engagement in HIV care at the clinic in which they enrolled in Mali. Strengths of the study include its large sample size and sophisticated analyses. However, overall enthusiasm for the study is weakened by time period of the data (2006-2013), which may not be as relevant today in the era of universal ART eligibility and the broader scale-up of HIV care that has occurred over time, and that the findings do not add particularly novel data to the current LTFU literature. Moreover, the lack of data on undocumented deaths and silent transfers (which the authors acknowledge) is a major limitation in understanding the factors associated with re-engagement in care for those who become lost, particularly given that >50% of the total study population became lost to follow-up using the authors’ definition.

Answer: Thank you for this comment. We did not have data for periods after 2015 because the “Ensemble pour une Solidarité Thérapeutique Hospitalière en Réseau” (ESTHER) initiative ended in December 2015. ESTHER, a French agency for AIDS in developing countries, provided since 2005 an electronic medical records system “Evaluation et Suivi Opérationnel des Programmes d’ESTHER” (ESOPE, Epiconcept, France) in order to monitor PLHIV care and effectiveness of ART. This system was gradually implemented in 16 care centres in Bamako and regions with a financial support from ESTHER. After ESTHER funding ended in December 2015, the global fund, the main funder of the Malian ART programme did not renew funding for the ESOPE database. We therefore do not have updated data after December 2015. We think that in a country which depends on external funds for the HIV care, it is important to show through our study the necessity of continuing the implementation of an electronic data system throughout the country for a better assessment of the quality of care and of the continuum of care. 

Abstract:

- There is no mention in the methods about using multiply imputation

Answer: We added the following sentence in the methods section of the Abstract: “Multiple imputation was used to deal with missing data”.

Introduction:

- The authors write “It would also useful to identify those among PLHIV LTFU after ART initiation who will subsequently re-engage in care to develop tailored strategies to improve linkage to care of PLHIV who initiate ART.” However, the recommendations given by the authors in the discussion / conclusion are only general policy recommendations. How do the authors believe that the findings in this study can specifically be used to address their statement in the introduction and have an impact on the current HIV epidemic?

Answer: We replaced the term “linkage” by “engagement” which is more appropriate. Also, we rewrote the discussion/conclusion section of the manuscript. We identified several individual and structural factors associated with less re-engagement in care such as advanced clinical stage and low CD4 count, starting ART in recent period, being followed in regional hospitals or regional outpatient clinics ≥ 5 km from home to care centre, and being treated more than 12 months with low gain in CD4 count during follow-up that could be modified by Malian national ART programme to improve engagement to care. We believe that our findings can be used by Malian national ART programme to improve the healthcare provision by increasing HIV screening to reduce late presentation to care, to enhance ART adherence with community-based adherence support intervention which may contribute to gain in CD4 count, and to decentralize provision of ART particularly in the regions to deal with transportation or programme growth issues. 

- “…jeopardize treatment efficacy and illness outcome, particularly by raising the risk of the acquisition of viral resistance mutations…” this statement could use a citation

Answer: We rewrote the introduction section and we removed the following sentence “…jeopardize treatment efficacy and illness outcome, particularly by raising the risk of the acquisition of viral resistance mutations…”.

Methods:

- The authors write “First, we assessed factors associated with LTFU by defining LTFU as an interval of at least six months without any clinical appointment for PLHIV not known to be dead or to have been transferred elsewhere.” Followed by “The date of LTFU was the date of the last centre visit plus three months (i.e. the date of the missed scheduled clinical visit) [4, 13].” These two sentences seem to contain contradictory definitions - the first LTFU definition is 6 months and the second is 3 months. The references cited do not help clarify this either, as the first reference (4) gives a summary of LTFU definitions and the second (13) used a definition of “loss to follow-up defined as a duration of >6 months between the last visit recorded and the closing date of the database”

Answer: We agree that our definition of LTFU may be confusing. We changed it as follows: 

“Loss to follow-up (LTFU) was defined as an interval of at least six months without any clinical appointment for PLHIV not known to be dead or to have been transferred elsewhere. The date of LTFU was the date of the first missed scheduled clinical visit [5, 12], as clinician’s first consider an individual to be LTFU at the time of the first missed visit.” 

However, MBeri et al [12, previously 13] used the threshold (≥ 180 days) proposed by Chi et al [4, previously 3] to build their definition. In Mberi et al study [12 previous 13], a patient was classified as LTFU if the patient had been followed up at least once after ART initiation, but had not had contact with the clinic for 180 days or more since their last recorded expected return date, or if there were 180 days or more between the expected date of return and the next clinic visit. A maximum of 3 months (90 days) of treatment can be dispensed to a patient at one time in Tshepang clinic. Therefore, according to our definition, patients that were classified as LTFU had not had treatment for 3 (if three months of treatment had been dispensed at the last clinic visit) to under 6 months (if less than 1 month of treatment had been dispensed to the patient at the last visit [12, previously 13]. Thus, Mberi et al did not assess LTFU before the database closure. We used the same definition of LTFU. 

Shepherd et al [5, previously 4] discussed the definition of LTFU proposed by Chi et al [4, previously 3]. They suggested to take into others components of the LTFU definition. According to Shepherd et al, if incidence of LTFU itself is the primary outcome, a reasonable choice could be a prospective definition of LTFU (i.e., 180 days, any encounters, prospective, time from last visit) counting any encounter (clinic or pharmacy) as a qualifying visit, with patients defined as lost at the time of their missed visit. From a clinician’s perspective, a patient is first considered lost at the time of his/her missed visit [5, previously 4]. We therefore took into account this suggestion from Shepherd et al [5, previously 4].

- Why were 5 km and 50 km used cutoffs for distance from the clinic?

Answer: For the cutoffs of the distance from home to care centre, the 5 km distance correspond to the decentralization policy which was set up in the country. We added the following sentence in statistical analyses section to clarify the choice of cutoffs for the distance: 

“In the effort to improve access to healthcare, the Malian government decentralized the outpatient clinics to get them closer (i.e. less than 5 km away) to the majority of the population. According to the third Malian Socio-Sanitary Development Programme (PRODESS III) 2014-2018, the population living less than 5 km from the outpatient clinic rose from 29% in 1998 to 56% in 2012 [14]. We therefore decided to divide the distance from home to care centre into 3 categories: short (< 5 km), intermediate ([5-50[ km) and long (≥ 50 km) distance to evaluate the impact of short and long distances on the subsequent re-engagement in care.”

- Time on ART is an important variable on care outcomes and is not included in the models

Answer: We agree with this comment and we included in our model the duration of ART until LTFU (months) divided into three categories (< 6, [6-12[ and ≥ 12 months). We found an interaction between duration of ART until LTFU and the estimation of the 12-month change in CD4 count, and we combined these two variables. Also, we observed that the Median (IQR) of 12-month change in CD4 count was null for PLHIV lost to follow-up before December 2013 who were treated for less than 6 months regardless the ‘return to care’ or ‘extended LTFU’ groups (see table at below). So, the two variables were combined into five categories as follows < 6 months, [6-12[ months and < 50 cells/µL, [6-12[ months and ≥ 50 cells/µL, ≥ 12 months and < 50 cells/µL, and ≥ 12 months and ≥ 50 cells/µL and take into account in our models.

Table. Median (IQR) of 12-month change in CD4 count by categories of duration of ART until LTFU (months) according to region and care centre expertise level (imputed data).

Median (IQR) of 12-month change in CD4 count (cells/µL) Categories of duration of ART until LTFU (months)

 < 6 [6-12[ ≥ 12

Return to care (n = 1,675) 0 (0-0)

(n=338) 79 (0-236)

(n=497) 99 (38-178)

(n=840)

Extended LTFU (n = 1,975) 0 (0-0)

(n=711) 0 (-9-62)

(n=384) 6 (-16-56)

(n=880)

Results:

- The tables contain densities of information and overwhelm the reader

- The % missing for each of the variables would be better listed in the tables or text as they were difficult to locate in the footnotes of such large tables

Answer: We added to our tables and in the text, the % of missing data for the variables 

which have missing data. 

- Presuming the tables show the results of the imputation analysis, there are no complete case analysis results shown or any comparison between the complete-case and imputation results

Answer: We do not perform the complete case analysis because we are going to lose nearly half of our study population regarding the high missing data (44%) for CD4 count at the time of loss to follow-up. 

- Table 1, WHO stage and CD4 count (cell/μL) row, Not LTFU column - does not sum to 100%

Answer: We agree that in previous Table 1, WHO stage and CD4 count (cell/μL) row, in the Not LTFU column - does not sum to 100%. We decided to focus on re-engagement in care and we removed from the manuscript previous Table 1 which showed factors associated with LTFU. 

Discussion:

- The first two paragraphs of the discussion recap results that have already been presented. It would be more informative to the reader to understand what the authors’ believe to be the significance of their findings.

- What would the authors believe to be cause of the observed variability in likelihood of re-engagement across the different time periods?

Answer: We rewrote the discussion section. We discussed the observed cumulative incidence rates of re-engagement in care in the second paragraph of the discussion section. 

- Imputation is risky if the imputed variables are not missing at random. Why do the authors believe that having a CD4 count, for example, is MAR and not also related to some of the same social-ecological factors that drive retention in care?

Answer: We added the following sentences in the statistical analyses section to explain why we believe that CD4 count is missing at random: 

“Multiple imputation is a valid approach for all missing at random mechanisms, whilst complete case analysis may give biased results when the chance of being a complete case depends on the observed values of the outcome (for example LTFU or death) [18]. When data are missing at random, any systematic differences between the observed and missing data can be explained by associations of the missing data with the observed data; for example if CD4 count measurement was more likely to be missing among PLHIV who did not re-engaged in care (‘extended LTFU’ group) but only because, as they were more likely to be in low socio-economic position, they were less likely to attend the clinical visit where CD4 count was measured [18]. Multiple imputation analyses would avoid bias only if enough variables predictive of missing values are included in the imputation model [15].”

Other general comments:

- The writing could use some grammatical editing. For example “LTFU” is defined as “loss to follow-up” however the phrase “PLHIV LTFU” is used repeatedly, and stating “people living with hiv loss-to-follow-up” doesn’t make sense grammatically. 

 Answer: we replaced in the manuscript all the terms “PLHIV LTFU” by “PLHIV lost to follow-up”.

---

## [Decision Letter · Decision Letter 1]

8 Apr 2020

PONE-D-19-19274R1

Re-engagement in care of people living with HIV lost to follow-up after initiation of antiretroviral therapy in Mali: who returns to care?

PLOS ONE

Dear Aliou BALDE,

Thank you for submitting your manuscript to PLOS ONE. After careful consideration, we feel that it has merit but does not fully meet PLOS ONE’s publication criteria as it currently stands. Therefore, we invite you to submit a revised version of the manuscript that addresses the points raised during the review process.

Specifically, you failed to address key points of one of the reviewers and this might revert the editorial decision if unsatisfactory.

We would appreciate receiving your revised manuscript by May 23 2020 11:59PM. To enhance the reproducibility of your results, we recommend that if applicable you deposit your laboratory protocols in protocols.io, where a protocol can be assigned its own identifier (DOI) such that it can be cited independently in the future. For instructions see: http://journals.plos.org/plosone/s/submission-guidelines#loc-laboratory-protocols

We look forward to receiving your revised manuscript.

Kind regards,

Joseph Fokam, Ph.D

Academic Editor

PLOS ONE

Reviewers' comments:

Reviewer's Responses to Questions

**Comments to the Author**

1. If the authors have adequately addressed your comments raised in a previous round of review and you feel that this manuscript is now acceptable for publication, you may indicate that here to bypass the “Comments to the Author” section, enter your conflict of interest statement in the “Confidential to Editor” section, and submit your "Accept" recommendation.

Reviewer #2: (No Response)

Reviewer #3: All comments have been addressed

2. Is the manuscript technically sound, and do the data support the conclusions?

Reviewer #2: Partly

Reviewer #3: Yes

3. Has the statistical analysis been performed appropriately and rigorously? 

Reviewer #2: I Don't Know

Reviewer #3: Yes

4. Have the authors made all data underlying the findings in their manuscript fully available?

Reviewer #2: No

Reviewer #3: (No Response)

5. Is the manuscript presented in an intelligible fashion and written in standard English?

Reviewer #2: No

Reviewer #3: Yes

6. Review Comments to the Author

Reviewer #2: Review report

The Authors did not address adequately to our comments therefore, We are maintaining the decision of major revision. The Authors did not provide adequate justification. This submission is replicating or derive from existing work. The Plos One Policy is clear for this case and Editor has to appreciate.

Authors have published results on LTFU in 2019: and they described LTFU in their papers: The current paper might be concentrated in re-engagement as primary outcome instead of repeating data on LTFU in the same population:

https://pubmed.ncbi.nlm.nih.gov/30270487-risk-factors-for-loss-to-follow-up-transfer-or-death-among-people-living-with-hiv-on-their-first-antiretroviral-therapy-regimen-in-mali/?from_term=Risk+factors+for+loss+to+follow+up%2C+transfer+or+death+among+people+living+on+their+first+antiretroviral+therapy+regimen+in++Mali&from_pos=1

The Authors should note that PLOS One publication criteria state that if a submitted study replicates or is very similar to previous work, authors must provide a sound scientific rationale for the submitted work and clearly reference and discuss the existing literature. Submissions that replicate or are derivative of existing work will likely be rejected if authors do not provide adequate justification:

(https://journals.plos.org/plosone/s/criteria-for-publication#loc-2)

Method:-How did they measure the socio-economic status? We did not found such data in the manuscript. They have reported Professional activity: Did they use that as socio-economic status?

The authors said that , they removed all the socioeconomic terms in their manuscript. Socio-economic terms are important in terms of controlling confounding. They have to provide rationale and how they have define and collect such variables instead of to remove. This is reduce the quality of their analysis.

In the method section: Authors should explain why they define periods of ART initiation in such period. Did they do in term of national guidelines?

Authors did not address to this comment: «To analyze the Re-engagement issue: How authors did measure the duration from LTFU and re-engagement: What was the end date since this is the outcome.» Readers will need to know How they measure the duration from LTFU and re-engagement. If the definition is not clear this can introduce bias and misclassification. For the patients who were definitively LTFU during the subsequent three-year follow-up period, How did Authors calculate the duration for them. The start date and End point are not clear: This still some confusion. If this is not clear; Authors should use logistic regression instead of Cox regression?

Authors said: «We assessed factors associated with re-engagement in care in PLHIV lost to follow-up. For this analysis, PLHIV who were lost to follow-up after December 31, 2013 were excluded to allow sufficient time to re-engage in care before the last database update (03/31/2015). » and «This study included all HIV-1-infected adults, aged ≥ 18 years, who began ART between January 1, 2006 and December 31, 2012 at one of the 16 care centres contributing to the ESOPE database, thus beginning treatment at least two years before the last database update (31 March 2015), and who returned for their one-month visit.» This can make confusion

Authors said: «We fitted linear regression, logistic regression, and multinomial regression models for CD4 counts at the start of ART and at LTFU (fourth root), WHO stage (1-2 or 3-4), and educational level (none, primary/koranic, secondary and higher), respectively. » We did not found any results reported to this analysis.

Authors said that they did not look for confounding variables. If this is the case, What will be the validity of such study: Did authors assess the internal validity of their study? During this process, authors should analyze type of errors which could occur in their study:

- Random errors( sample size and sampling procedure)

- Selection Bias (LFU in the RCT)

- Information Bias: How the data was measured; (observer, instrument, observed and environment) . It can be : error measurement, improper diagnostic criteria, omissions, imprecise measurement. To assess it we use sensitivity and specificity to determine the degree of misclassification ( That why we need clear of how they measure the duration from LTFU and re-engagement)

- Confounding: we can control it in two stage:

in design: by randomization, stratification, matching (case-control) and restriction( by inclusion and exclusion criteria )

in Analysis: Stratification or by using multivariate technique

Authors assessed the interactions between several variables without providing the reason of doing that.

Confounding factors is what looks like a causal relationship between a supposed factor and an outcome( disease, other event,..) may be due to another factor not taken into consideration.

Manuscripts submitted to PLOS ONE are expected to report statistical methods in sufficient detail for others to replicate the analysis performed. Ensure that results are rigorously reported in accordance with community standards and that the statistical methods employed are appropriate for the study design.

I would request that Authors put publically their dataset for next version.

Authors said : «The study was approved by the national AIDS programme in Mali (approval letter available). …, Written or oral informed consent was obtained from each PLHIV before medical recording.» Is This IRB committee? And Why Written and oral consent? Is this what the IRB committee required?

Authors are using ESOPE database: So are they using secondary data? If so, They should state it.

What is the rationale to classify age as follows: < 30 years, 30-40 years and ≥ 40 years? ( Why not 18-24, 25-35,….). You can notice that authors have to provide the reason. Authors should notice that this way can explain residual confounding.

What is the rational to present Marital status data as polygamous, monogamous and others. We would like to know how never married, divorced and widowed is distributed. And How was the issue per each category.

How many patients were in ESOPE database from January 1, 2006 and December 31, 2012. How many were LTFU during the same period? This is not clear

Authors said : «Of the 8,307 HIV-positive adults contributing to the ESOPE database and starting ART between 2006 and 2012 with CD4 or WHO stage data available at the start of ART, 7,975 individuals returned for their one-month visit and were included in this study (Fig. 2). The individual characteristics including missing» 8,307 were previously LTFU?

I suggest to Authors for key co-variates ( significant in cox model) to present them in Kaplan Meier curve: This will be easy to see the difference.

Authors remove the Conclusion section but maintained in the abstract?

Conclusion: MAJOR REVISION

Reviewer #3: The authors have adequately addressed my comments and the manuscript has been significantly improved.

7. PLOS authors have the option to publish the peer review history of their article (what does this mean?). If published, this will include your full peer review and any attached files.

Reviewer #2: No

Reviewer #3: No

---

## [Author Response · Author response to Decision Letter 1]

29 May 2020

Review Comments to the Author

Reviewer #2: Review report

The Authors did not address adequately to our comments therefore, We are maintaining the decision of major revision. The Authors did not provide adequate justification. This submission is replicating or derive from existing work. The Plos One Policy is clear for this case and Editor has to appreciate.

Authors have published results on LTFU in 2019: and they described LTFU in their papers: The current paper might be concentrated in re-engagement as primary outcome instead of repeating data on LTFU in the same population:

https://pubmed.ncbi.nlm.nih.gov/30270487-risk-factors-for-loss-to-follow-up-transfer-or-death-among-people-living-with-hiv-on-their-first-antiretroviral-therapy-regimen-in-mali/?from_term=Risk+factors+for+loss+to+follow+up%2C+transfer+or+death+among+people+living+on+their+first+antiretroviral+therapy+regimen+in++Mali&from_pos=1

The Authors should note that PLOS One publication criteria state that if a submitted study replicates or is very similar to previous work, authors must provide a sound scientific rationale for the submitted work and clearly reference and discuss the existing literature. Submissions that replicate or are derivative of existing work will likely be rejected if authors do not provide adequate justification:

(https://journals.plos.org/plosone/s/criteria-for-publication#loc-2)

Answer: We are also against duplicate publications. However, we do not think that this is the case for the reasons provided below. Factors associated with re-engagement in care are not necessary the same as factors associated with LTFU.

We conducted our first study in order to estimate the cumulative incidence of loss of follow-up and to assess risk factors for this event in PLVIH in Mali. We had observed that some of PLHIV lost to follow-up subsequently returned to care with sometimes more than one care interruption during their follow-up and that some did not. Given that Mali, a resource-limited country, does not have an electronic national database, death registries or resources dedicated to active tracing, we wanted to identify among PLHIV with a first interruption of care factors associated with a subsequent return to care. 

In the resubmitted version, we had already removed from the main text all references to the description of the general population, but this population was described in supplementary material and we referred to the supplementary material in the text. We have now completely deleted this part from the main text in the result section and we removed the supplementary material, and we concentrated in re-engagement in care only.

The first two paragraphs of the result section were changed from:

" Of the 8,307 HIV-positive adults contributing to the ESOPE database and starting ART between 2006 and 2012 with CD4 or WHO stage data available at the start of ART, 7,975 individuals returned for their one-month visit and were included in this study (Fig. 2). The individual characteristics including missing data are described in Supplementary Table 1 based on their subsequent follow-up status (‘Remained in care’, ‘LTFU before 12/31/2013’, ‘LTFU after 12/31/2013’, ‘Transfer’ or ‘Death’). The percentages of missing data were 14% for CD4 cell count, 9% for WHO stage, and 15% for educational level. During the study period, 2,302 (29%) PLHIV remained in care, 3,650 (46%) and 1,008 (12%) were lost to follow-up for at least six months during their follow-up before and after 12/31/2013, respectively. Also, during the study period, 595 (8%) and 420 (5%) were known to be transferred to another care centre and to be dead, respectively (S1 Table). 

The study included 5,330 (67%) women and median (interquartile range [IQR]) age was 36 (29-43). The most prescribed initial ART regimen contained zidovudine (AZT) or stavudine (d4T) and lamivudine (3TC) plus nevirapine (NVP) or efavirenz (EFV) for 6,106 (77%) individuals. The second most prescribed one contained tenofovir (TDF) and emtricitabine (FTC) or 3TC plus NVP or EFV for 1,332 (17%) individuals. Overall median follow-up was 26 months (IQR, 8-48). Characteristics at the start of ART of the individuals were slightly different according to the region of care and the care centre expertise level (S2 Table)."

To: 

" Of the 8,037 HIV-positive adults contributing to the ESOPE database and starting ART between 2006 and 2012 with CD4 or WHO stage data available at the start of ART, 7,975 individuals returned for their one-month visit and 4,658 were lost to follow-up before the database closure (03/31/2015) (Fig. 2). Among those PLHIV lost to follow-up, 3,650 were lost to follow-up before 12/31/2013 and were included in this study (Fig. 2). The study included 2,380 (65%) women and median (interquartile range [IQR]) age was 35 (29-43). The most prescribed initial ART regimen contained zidovudine (AZT) or stavudine (d4T) and lamivudine (3TC) plus nevirapine (NVP) or efavirenz (EFV) for 2,958 (81%) individuals. The second most prescribed one contained tenofovir (TDF) and emtricitabine (FTC) or 3TC plus NVP or EFV for 429 (12%) individuals. Overall median duration of ART before LTFU was 11 months (IQR, 5-22) (S1 Table).".

Methods : How did they measure the socio-economic status? We did not found such data in the manuscript. They have reported Professional activity: Did they use that as socio-economic status?

The authors said that, they removed all the socioeconomic terms in their manuscript. Socio-economic terms are important in terms of controlling confounding. They have to provide rationale and how they have define and collect such variables instead of to remove. This is reduce the quality of their analysis.

Answer: As we previously stated, we used the occupation (Professional activity) to measure the socioeconomic status. We do not have income data in the ESOPE database. In the absence of income data, the more conventional indicators of socioeconomic status are completed schooling and occupation (Duncan GJ et al, Am J Public Health 2002; Mackenbach JP et al, N Engl J of Med 2008). Education is the most stable measure of socioeconomic position, because it is normally completed early in adulthood, avoiding most of the problems of reverse causation (Mackenbach JP et al, Lancet Public Health 2019). The educational level collected in ESOPE database is the “The last attained educational level”. However, as suggested we kept the occupation in our final model and changed our results in the paper (see results in the Table 1 below). 

We changed the sentence in the statistical analyses section, from:

"The considered individual characteristics at ART initiation were a combined variable of sex and pregnancy, age, WHO stage and CD4 count, period of ART initiation, marital status, last attained educational level and a variable combining the location (Bamako or one of the regions), expertise level (outpatient clinic or hospital) of the care centre and its distance from the individual’s home."

To

"The considered individual characteristics at ART initiation were a combined variable of sex and pregnancy, age, WHO stage and CD4 count, period of ART initiation, marital status, last attained educational level, professional activity and a variable combining the location (Bamako or one of the regions), expertise level (outpatient clinic or hospital) of the care centre and its distance from the individual’s home."

We also added the following sentence in the statistical analyses section: 

“Professional activity categories were obtained by combining types and sectors of employment, housewife without any professional activity, public or private sector salaried employment, various non-salaried activities, such as being self-employed, farmer/fisherman, other (unemployed including students) and missing.”

Of note, we previously mentioned in our discussion that in several studies from sub-Saharan Africa, PLHIV lost to follow-up reported that poverty and the lack of money for transportation cost were barriers to return to care [7, 37-39].

Table 1. Factors associated with re-engagement in care in the 36 months after LTFU among people living with HIV starting ART in Mali subsequent to LTFU (Cox model with imputed data)

Characteristics Extended LTFU

N = 1,975 Return to care

N = 1,675 

Univariable 

Multivariable

 n (%) n (%) HR (95% CI) P HR (95% CI) P

Sex and pregnancy

Men

Non-pregnant women

Pregnant women 

723 (36.6)

1,171 (59.3)

81 (4.1) 

547 (32.7)

926 (55.3)

202 (12.0) 

1

1.04 (0.94-1.15)

2.10 (1.81-2.43) < 0.0001 

1

1.04 (0.91-1.19)

1.56 (1.30-1.86) < 0.0001

Age (years)

< 30

[30-40[

≥ 40 

551 (27.9)

761 (38.5)

663 (33.6) 

485 (29.0)

673 (40.2)

517 (30.8) 

1

1.02 (0.91-1.14)

0.92 (0.82-1.04) 0.1980 

1

1.10 (0.98-1.24)

1.04 (0.91-1.19) 0.1502

WHO stage and CD4 count (cells/µL) (missing WHO stage, 9%; missing CD4, 14%)

WHO stage 3-4 and CD4 < 200

WHO stage 3-4 and CD4 ≥ 200

WHO stage 1-2 and CD4 < 200

WHO stage 1-2 and CD4 ≥ 200 

662 (33.5)

274 (13.9)

612 (31.0)

427 (21.6) 

404 (24.1)

208 (12.4)

573 (34.2)

490 (29.3) 

1

1.19 (0.99-1.43)

1.33 (1.17-1.52)

1.54 (1.35-1.75) < 0.0001 

1

1.08 (0.90-1.30)

1.12 (0.98-1.28)

1.23 (1.07-1.41) 0.0102

Periods of ART initiation

2006-2007

2008-2009

2010-2012 

598 (30.3)

391 (19.8)

986 (49.9) 

817 (48.8)

479 (28.6)

379 (22.6) 

1

0.96 (0.87-1.07)

0.41 (0.36-0.46) < 0.0001 

1

1.20 (1.06-1.36)

0.56 (0.49-0.64) < 0.0001

Marital status

Monogamousa 

Polygamous

Single

Widowed

Divorced

Missing 

869 (44.0)

358 (18.1)

251 (12.7)

221 (11.2)

97 (4.9)

179 (9.1) 

699 (41.7)

353 (21.1)

226 (13.5)

240 (14.3)

97 (5.8)

60 (3.6) 

1

1.14 (1.01-1.29)

1.09 (0.94-1.25)

1.22 (1.06-1.40)

1.17 (0.95-1.43)

0.50 (0.38-0.64) <0.0001 

1

1.06 (0.94-1.20)

0.99 (0.85-1.15)

1.08 (0.94-1.25)

1.04 (0.85-1.28)

0.57 (0.43-0.77) 0.0094

Educational level (missing, 15%)

None

Primary/Koranic

Secondary

Higher 

994 (50.3)

578 (29.2)

289 (14.7)

114 (5.8) 

759 (45.3)

525 (31.3)

286 (17.1)

105 (6.3) 

1

1.13 (1.00-1.28)

1.19 (1.03-1.38)

1.13 (0.90-1.42) 0.0283 

1

1.03 (0.91-1.17)

1.12 (0.97-1.30)

1.04 (0.81-1.34) 0.1779

Professional activityb

Housewife

Public sector staff

Private sector staff

Self-employed

Farmer/Fisherman

Other

Missing 

1,435 (39.3)

209 (5.7)

691 (18.9)

521 (14.3)

243 (6.7)

433 (11.9)

118 (3.2) 

768 (38.9)

104 (5.3)

341 (17.3)

287 (14.5)

164 (8.3)

223 (11.3)

88 (4.4) 

1

1.09 (0.90-1.33)

1.11 (0.98-1.25)

0.96 (0.84-1.11)

0.63 (0.50-0.79)

1.06 (0.91-1.22)

0.49 (0.34-0.70) <0.0001 

1

1.02 (0.82-1.27)

1.04 (0.90-1.20)

1.09 (0.92-1.29)

0.78 (0.61-1.00)

0.98 (0.83-1.16)

1.17 (0.81-1.16) 0.1477

Regions, care centre & distance (km)

Bamako outpatient clinics

Bamako hospitals & < 5

Bamako hospitals & [5-50[

Bamako hospitals & ≥ 50

Regional outpatient clinics & < 5

Regional outpatient clinics & [5-50[

Regional outpatient clinics & ≥ 50

Regional hospitals & [5-50[

Regional hospitals & ≥ 50 

834 (42.2)

51 (2.6)

349 (17.7)

52 (2.6)

95 (4.8)

74 (3.8)

37 (1.9)

355 (17.9)

128 (6.5) 

1,005 (60.0)

68 (4.0)

300 (17.9)

55 (3.3)

96 (5.7)

38 (2.3)

21 (1.3)

71 (4.2)

21 (1.3) 

1

1.08 (0.86-1.36)

0.82 (0.73-0.93)

0.97 (0.74-1.26)

0.93 (0.76-1.14)

0.53 (0.39-0.72)

0.58 (0.38-0.88)

0.25 (0.19-0.31)

0.20 (0.13-0.31) < 0.0001 

1

1.22 (0.97-1.54)

0.91 (0.79-1.04)

1.18 (0.92-1.52)

0.96 (0.74-1.23)

0.78 (0.56-1.09)

0.64 (0.41-1.01)

0.36 (0.28-0.47)

0.33 (0.21-0.51) < 0.0001

Duration of ART until LTFU (month) and 12-month change in CD4 count (cell/µL)c (missing CD4 at LTFU, 44%)

< 6

[6-12[ and < 50

[6-12[ and ≥ 50

≥ 12 and < 50

≥ 12 and ≥ 50 

711 (36.0)

283 (14.3)

101 (5.1)

641 (32.5)

239 (12.1) 

338 (20.2)

227 (13.6)

270 (16.1)

250 (14.9)

590 (35.2) 

1

1.50 (1.26-1.77)

3.15 (2.66-3.72)

0.90 (0.75-1.08)

3.09 (2.70-3.53) < 0.0001 

1

1.46 (1.23-1.73)

2.61 (2.21-3.08)

0.87 (0.72-1.04)

2.43 (2.12-2.78) < 0.0001

Abbreviations: LTFU, loss to follow-up; ART, antiretroviral therapy; HR, hazards ratio; CI, confidence interval. aMarried or living with a partner. bProfessional activities: professional activity categories combined types and sectors of employment, housewife without any professional activity, public or private sector salaried employment, various non-salaried activities, such as being self-employed, farmer/fisherman, and other (unemployed including students). cVariable combining the duration of ART until LTFU (month) and the estimation of the 12-month change in CD4 count (cells/ µL).

In the method section: Authors should explain why they define periods of ART initiation in such period. Did they do in term of national guidelines?

Answer: Yes, the periods that we used correspond to the changes of the national guidelines. To clarify, we added the following sentences (underlined) to the study population section: 

“At the end of 2005, first national guidelines on HIV care and ART were released, with subsequent revisions in 2008, 2010 and 2013 according to WHO guidelines. In 2005, guidelines recommended treatment for all PLHIV with CD4 counts ≤ 200 cells/µL, at WHO stage 3 and CD4 counts ≤ 350 cells/µL, or at WHO stage 4. In 2008 and 2010, all PLHIV with CD4 counts ≤ 350 cells/µL or at WHO stage 3-4 had to be treated. Pregnant women beginning ART at WHO stage 1-2 and/or CD4 counts > 350 cells/µL were excluded from this study, because ART initiated for pregnancy was stopped after delivery if the clinical stage and biological results at the time of treatment initiation were not in the range of the indication for treatment in the national guidelines for ART [9]. 

In Mali, the recommended initial ART regimen for HIV-1-infected individuals consists of two nucleoside reverse transcriptase inhibitors (NRTI), including lamivudine or emtricitabine, plus one non-nucleoside reverse transcriptase inhibitor (NNRTI), nevirapine or efavirenz. In 2010, stavudine was removed from the recommended initial ART regimen. The currently recommended initial ART regimen is tenofovir plus lamivudine and efavirenz [9].”

 Authors did not address to this comment: «To analyze the Re-engagement issue: How authors did measure the duration from LTFU and re-engagement: What was the end date since this is the outcome.» Readers will need to know How they measure the duration from LTFU and re-engagement. If the definition is not clear this can introduce bias and misclassification. For the patients who were definitively LTFU during the subsequent three-year follow-up period, How did Authors calculate the duration for them. The start date and End point are not clear: This still some confusion. If this is not clear; Authors should use logistic regression instead of Cox regression?

Answer: This was explained but perhaps likely not very clearly. To be clearer, we corrected statistical analyses section as follows: 

The sentence was:

" We assessed factors associated with re-engagement in care in PLHIV lost to follow-up. For this analysis, PLHIV who were lost to follow-up after December 31, 2013 were excluded to allow sufficient time to re-engage in care before the last database update (03/31/2015). The baseline for this analysis was the date of LTFU. Follow-up time was censored at closure of the database or three years after the date of LTFU, whichever occurred first. Factors associated with re-engagement in care were assessed using univariable and multivariable Cox models [13] allowing the comparison of PLHIV who returned to care after LTFU (‘return to care’ group) and those who did not return to care after LTFU (‘extended LTFU’ group). "

We changed it to:

“We estimated the cumulative incidence of re-engagement in care and assessed factors associated with this event in PLHIV lost to follow-up. The baseline for this analysis was the date of LTFU. The time without follow-up was counted from the date of LTFU to the date of re-engagement in care, three years after the date of LTFU or closure of the database (03/31/2015), whichever occurred first. The cumulative incidence of re-engagement in care was estimated using Kaplan-Meier. Factors associated with re-engagement in care were assessed using univariable and multivariable Cox models [13] allowing the comparison of PLHIV who returned to care after LTFU (‘return to care’ group) and those who did not return to care after LTFU (‘extended LTFU’ group).”

Authors said: «We assessed factors associated with re-engagement in care in PLHIV lost to follow-up. For this analysis, PLHIV who were lost to follow-up after December 31, 2013 were excluded to allow sufficient time to re-engage in care before the last database update (03/31/2015). » and «This study included all HIV-1-infected adults, aged ≥ 18 years, who began ART between January 1, 2006 and December 31, 2012 at one of the 16 care centres contributing to the ESOPE database, thus beginning treatment at least two years before the last database update (31 March 2015), and who returned for their one-month visit.» This can make confusion

Answer: We agree with this comment and we corrected the study population section as follows: 

From:

"This study included all HIV-1-infected adults, aged ≥ 18 years, who began ART between January 1, 2006 and December 31, 2012 at one of the 16 care centres contributing to the ESOPE database, thus beginning treatment at least two years before the last database update (31 March 2015), and who returned for their one-month visit."

To:

“This study included all PLHIV who were lost to-follow-up before December 31, 2013, to allow sufficient time to re-engage in care before the last database update (03/31/2015). These PLHIV lost to follow-up were selected from all HIV-1-infected adults, aged ≥ 18 years, who began ART between January 1, 2006 and December 31, 2012 at one of the 16 care centres contributing to the ESOPE database, thus beginning treatment at least two years before the last database update (31 March 2015), and who returned for their one-month visit.”

Authors said: «We fitted linear regression, logistic regression, and multinomial regression models for CD4 counts at the start of ART and at LTFU (fourth root), WHO stage (1-2 or 3-4), and educational level (none, primary/koranic, secondary and higher), respectively. » We did not find any results reported to this analysis.

Answer: This sentence refers to the method used to calculate missing values for PLHIV for whom some data were missing. Then all PLHIV are analyzed and we provided the analyze for all PLHIV. 

Authors said that they did not look for confounding variables. If this is the case, What will be the validity of such study: Did authors assess the internal validity of their study? During this process, authors should analyze type of errors which could occur in their study:

- Random errors (sample size and sampling procedure)

- Selection Bias (LFU in the RCT)

- Information Bias: How the data was measured; (observer, instrument, observed and environment). It can be: error measurement, improper diagnostic criteria, omissions, imprecise measurement. To assess it we use sensitivity and specificity to determine the degree of misclassification (That why we need clear of how they measure the duration from LTFU and re-engagement)

- Confounding: we can control it in two stage:

in design: by randomization, stratification, matching (case-control) and restriction (by inclusion and exclusion criteria)

in Analysis: Stratification or by using multivariate technique

Answer: We are sorry we were not clear in what we wanted to say. We wanted to say that the study did not focus specifically on identifying the confounders. We assessed in multivariable analyses the association between re-engagement in care and the characteristics of PLHIV lost to follow-up among which potential confounding variables. We thus adjusted on the potential confounders that were available in the database. If we consider social variables, we think that depression and income could be confounding factors for marital status or professional activity, respectively, but we did not have these data. More, we looked for interactions between the variables and took into account the interactions we found in our model. We also checked for multicollinearity. We conducted a multivariable analysis taking into account the relation between potential confounders and re-engagement in care.

Authors assessed the interactions between several variables without providing the reason of doing that. 

Answer: We now provide the reasons for investigating these interactions. The end of the first paragraph of the statistical section was changed from:

" We assessed the interaction between the duration of ART until LTFU and the estimation of the 12-month change in CD4 counts before LTFU, and the interactions between region of care, care centre expertise level and distance from home to care centre. We also assessed the interactions between the period of ART initiation and the following variables: Sex and pregnancy, age, WHO stage and CD4 count, and distance from home to care centre."

To:

“We assessed the interaction between the duration of ART until LTFU and the estimation of the 12-month change in CD4 counts before LTFU because the change in CD4 usually depends on the time on ART. We assessed the interactions between region of care, care centre expertise level and distance from home to care centre because recruitment of PLHIV can be different between the different centres and regions according to distance from home. Given the revisions of national guidelines on HIV care and ART, we also assessed the interactions between the period of ART initiation and the following variables: Sex and pregnancy, age, WHO stage and CD4 count, and distance from home to care centre.”

Confounding factors is what looks like a causal relationship between a supposed factor and an outcome (disease, other event,..) may be due to another factor not taken into consideration.

Manuscripts submitted to PLOS ONE are expected to report statistical methods in sufficient detail for others to replicate the analysis performed. Ensure that results are rigorously reported in accordance with community standards and that the statistical methods employed are appropriate for the study design.

I would request that Authors put publically their dataset for next version.

Answer: Our dataset can be made available upon request.

Authors said : «The study was approved by the national AIDS programme in Mali (approval letter available). …, Written or oral informed consent was obtained from each PLHIV before medical recording.» Is This IRB committee? And Why Written and oral consent? Is this what the IRB committee required?

Answer: The national AIDS program is the competent Institutional Review Board (IRB)for a study on PLHIV. Written and oral consent from any new HIV-positive individual is what the IRB committee requires before the creation of a paper or electronic medical record.

Authors are using ESOPE database: So are they using secondary data? If so, They should state it.

Answer: We used data from the ESOPE database. We do not know what are secondary data.

What is the rationale to classify age as follows: < 30 years, 30-40 years and ≥ 40 years? ( Why not 18-24, 25-35,….). You can notice that authors have to provide the reason. Authors should notice that this way can explain residual confounding.

Answer: We added the following sentence in the statistical analyses section:

“The age at ART initiation was divided into three categories using terciles.”

What is the rational to present Marital status data as polygamous, monogamous and others. We would like to know how never married, divorced and widowed is distributed. And How was the issue per each category.

Answer: We aimed to have a reasonably small number of categories for the variables. The marital status was divided into three categories (monogamous or living with a partner, polygamous and other (living alone -single, widowed, divorced- and few missing data)).

We provided the distribution of other categories (single, widowed, divorced and missing) of marital status in Table 1. Also, in the table below, we show the different categories of marital status (monogamous, polygamous, single, widowed, divorced) were not associated with subsequent re-engagement in care. 

We added a sentence in the result section (second paragraph of 'factors associated with re-engagement in care’):

“Conversely, PLHIV who started ART in 2010-2012 (aHR 0.56; 95%CI 0.49-0.64) were less likely to re-engage in care than those initiating ART in 2006-2007 (Table 1). Marital status and professional activity were not associated with re-engagement in care, except farmers/fishermen who were at lower risk of re-engagement in care compared to housewives. 

How many patients were in ESOPE database from January 1, 2006 and December 31, 2012. How many were LTFU during the same period? This is not clear

Authors said : «Of the 8,307 HIV-positive adults contributing to the ESOPE database and starting ART between 2006 and 2012 with CD4 or WHO stage data available at the start of ART, 7,975 individuals returned for their one-month visit and were included in this study (Fig. 2). The individual characteristics including missing» 8,307 were previously LTFU?

Answer: There was an error on the number, it is 8,037 instead of 8,307. 

We corrected the section.

I suggest to Authors for key co-variates (significant in cox model) to present them in Kaplan Meier curve: This will be easy to see the difference.

Answer: We do not think this is a good idea because Kaplan Meier only gives the univariable results of the association between the covariates and return to care, without adjustment on confounders.

Authors remove the Conclusion section but maintained in the abstract?

Answer: We did not remove the conclusion in discussion section. In this conclusion we mentioned the factors associated with re-engagement in care and formulated recommendations for the Malian AIDS programme. We just removed the interpretations from the conclusion of the abstract. 

Conclusion: MAJOR REVISION

---

## [Decision Letter · Decision Letter 2]

31 Jul 2020

PONE-D-19-19274R2

Re-engagement in care of people living with HIV lost to follow-up after initiation of antiretroviral therapy in Mali: who returns to care?

PLOS ONE

Dear Dr. BALDE,

Thank you for submitting your manuscript to PLOS ONE. After careful consideration, we feel that it has merit but does not fully meet PLOS ONE’s publication criteria as it currently stands. Therefore, we invite you to submit a revised version of the manuscript that addresses the points raised during the review process.

We look forward to receiving your revised manuscript.

Kind regards,

Joseph Fokam, Ph.D

Academic Editor

PLOS ONE

Reviewers' comments:

Reviewer's Responses to Questions

**Comments to the Author**

1. If the authors have adequately addressed your comments raised in a previous round of review and you feel that this manuscript is now acceptable for publication, you may indicate that here to bypass the “Comments to the Author” section, enter your conflict of interest statement in the “Confidential to Editor” section, and submit your "Accept" recommendation.

Reviewer #3: All comments have been addressed

2. Is the manuscript technically sound, and do the data support the conclusions?

Reviewer #3: Yes

3. Has the statistical analysis been performed appropriately and rigorously? 

Reviewer #3: (No Response)

4. Have the authors made all data underlying the findings in their manuscript fully available?

Reviewer #3: Yes

5. Is the manuscript presented in an intelligible fashion and written in standard English?

Reviewer #3: Yes

6. Review Comments to the Author

Reviewer #3: My only suggestion would be to clarify the definition of LTFU under the "Definitions" paragraph in the methods. This currently states "Loss to follow-up (LTFU) was defined as an interval of at least six months without any clinical appointment for PLHIV not known to be dead or to have been transferred elsewhere". I presume the authors mean "attended visit" -- the term 'visit' is used in the remainder of this paragraph and it would be better to keep it consistent.

7. PLOS authors have the option to publish the peer review history of their article (what does this mean?). If published, this will include your full peer review and any attached files.

Reviewer #3: No

---

## [Author Response · Author response to Decision Letter 2]

31 Jul 2020

Review Comments to the Author

Reviewer #3: My only suggestion would be to clarify the definition of LTFU under the "Definitions" paragraph in the methods. This currently states "Loss to follow-up (LTFU) was defined as an interval of at least six months without any clinical appointment for PLHIV not known to be dead or to have been transferred elsewhere". I presume the authors mean "attended visit" -- the term 'visit' is used in the remainder of this paragraph and it would be better to keep it consistent.

Answer: To remain consistent, we corrected the first paragraph of “Definitions” in the methods section as follows: 

From:

"Loss to follow-up (LTFU) was defined as an interval of at least six months without any clinical appointment for PLHIV not known to be dead or to have been transferred elsewhere. The date of LTFU was the date of the first missed scheduled clinical visit [5, 12], as clinician’s first consider an individual to be LTFU at the time of the first missed visit."

To:

"Loss to follow-up (LTFU) was defined as an interval of at least six months without any clinical visit for PLHIV not known to be dead or to have been transferred elsewhere. The date of LTFU was the date of the first missed scheduled clinical visit [5, 12], as clinician’s first consider an individual to be LTFU at the time of the first missed visit."

---

## [Editor Report · Decision Letter 3]

24 Aug 2020

Re-engagement in care of people living with HIV lost to follow-up after initiation of antiretroviral therapy in Mali: who returns to care?

PONE-D-19-19274R3

Dear Dr. BALDE,

We’re pleased to inform you that your manuscript has been judged scientifically suitable for publication and will be formally accepted for publication once it meets all outstanding technical requirements.

Kind regards,

Joseph Fokam, Ph.D

Academic Editor

PLOS ONE
---

## [Editor Report · Acceptance letter]

25 Aug 2020

PONE-D-19-19274R3 

Re-engagement in care of people living with HIV lost to follow-up after initiation of antiretroviral therapy in Mali: who returns to care? 

Dear Dr. Baldé:

I'm pleased to inform you that your manuscript has been deemed suitable for publication in PLOS ONE. Congratulations! Your manuscript is now with our production department. 

Kind regards, 

on behalf of

Dr. Joseph Fokam 

Academic Editor

PLOS ONE